# Endothelial cell-derived CD95 ligand serves as a chemokine in induction of neutrophil slow rolling and adhesion

Liang Gao[1], Gülce Sila Gülcüler[1], Lieke Golbach[2], Helena Block[2], Alexander Zarbock[2], Ana Martin-Villalba[1]*

[1]Division of Molecular Neurobiology, German Cancer Research Center, Heidelberg, Germany; [2]Department of Anesthesiology and Critical Care Medicine, University of Münster, Münster, Germany

**Abstract** Integrin activation is crucial for the regulation of leukocyte rolling, adhesion and trans-vessel migration during inflammation and occurs by engagement of myeloid cells through factors presented by inflamed vessels. However, endothelial-dependent mechanisms of myeloid cell recruitment are not fully understood. Here we show using an autoperfused flow chamber assay of whole blood neutrophils and intravital microscopy of the inflamed cremaster muscle that CD95 mediates leukocyte slow rolling, adhesion and transmigration upon binding of CD95-ligand (CD95L) that is presented by endothelial cells. In myeloid cells, CD95 triggers activation of Syk-Btk/PLCγ2/Rap1 signaling that ultimately leads to integrin activation. Excitingly, CD95-deficient myeloid cells exhibit impaired bacterial clearance in an animal model of sepsis induced by cecal ligation and puncture (CLP). Our data identify the cellular and molecular mechanisms underlying the chemoattractant effect of endothelial cell-derived CD95L in induction of neutrophil recruitment and support the use of therapeutic inhibition of CD95's activity in inflammatory diseases.

*For correspondence: a.martin-villalba@dkfz.de

## Introduction

Leukocyte recruitment comprises of a cascade with four major steps: slow rolling, firm adhesion, intraluminal crawling and trans-vessel migration (*Ley et al., 2007*). Slow rolling and firm adhesion are mediated by selectin- and chemokine-induced integrin signaling. Selectin is expressed and presented to the vessel lumen by inflamed endothelial cells (*Zarbock et al., 2011*). E-selectin engagement with PSGL-1 and CD44 ligands induces activation of the Src family kinases (SFKs) Hck, Fgr and Lyn (*Yago et al., 2010*) which then phosphorylate and activate immunoreceptor tyrosine-based activation motif (ITAM)-bearing adaptor protein Fc receptor common γ signaling chain (FcRγ) and DNAX activation protein of 12 kDa (DAP12) (*Zarbock et al., 2008*). These activated adaptor proteins recruit and phosphorylate spleen tyrosine kinase (Syk), which in turn activates Bruton's tyrosine kinase (Btk) (*Mueller et al., 2010*; *Yago et al., 2010*). Btk further activates the phosphoinositide 3-kinase (PI3K), phospholipase C γ2 (PLCγ2) and p38 mitogen-activated protein kinase (p38 MAPK) pathways that mediate the integrin signaling to induce slow rolling.

E-selectin- and integrin-mediated rolling crucially depend on Syk activation via binding to phosphorylated ITAM-domains (*Mueller et al., 2010*; *Yago et al., 2010*; *Zarbock et al., 2007*). Interestingly, an ITAM-like motif was identified as a docking site for Src homology domain 2 (SH2)-containing proteins in CD95 of neutrophils (*Daigle et al., 2002*). In CD95L-stimulated myeloid cells, we identified Lyn as the major SFK that phosphorylates CD95's tyrosine, thereby allowing recruitment of Syk, which via the PI3K/MMP9 pathway results in myeloid cell migration to the inflammatory

**eLife digest** When tissues are damaged or infected, the body produces an inflammatory response. Neutrophils – a type of white blood cell – play an important part in this response. These cells normally circulate through the bloodstream, and are recruited to the inflamed site by chemical signals sent out by immune cells in the damaged tissue. This causes passing neutrophils to migrate through the wall of the blood vessel to gain access to the inflamed tissue.

The neutrophils go through a sequence of steps before they can pass through the blood vessel wall. After initially tethering to the cells that line the blood vessel, the neutrophils experience a period of "slow rolling" across the vessel lining, before tightly adhering to one of the cells.

In 2010, researchers determined that a protein on the neutrophil's surface, known as CD95, helps the cell migrate through blood vessel walls. This protein interacts with a "ligand" molecule on the surface of the cells that line the blood vessel. However, it remains unclear whether CD95 and its ligand play a role in the steps that lead up to the neutrophils migrating through the blood vessel wall.

Gao et al. – who include researchers involved in the 2010 study – now show that activating CD95 in neutrophils also triggers the cell's slow rolling and adhesion. Experiments performed on mouse cells and tissues showed that the cells that line the blood vessels present the CD95 ligand on their surfaces in order to activate CD95 in the neutrophils circulating in the bloodstream. This ultimately leads to neutrophil slow rolling and adhesion. Further experiments in mice showed that this ability of CD95 to recruit neutrophils to inflamed sites was crucial for clearing bacteria in cases of sepsis, where infection causes the immune system to damage the body's own tissues.

Future studies could address whether inhibiting CD95's activity could help to treat diseases that feature uncontrolled white blood cell recruitment, including various cancers and autoimmune diseases.

site (*Letellier et al., 2010*). However, the roles of CD95 in the initial cellular processes of myeloid cell recruitment, such as rolling and adhesion, remain unknown.

CD95 (Fas/Apo-1) was initially described as a death receptor mediating apoptosis via formation of the death-inducing signal complex (DISC) which further leads to activation of downstream caspases and apoptosis (*Peter et al., 2007*). Interestingly, in T cells the apoptotic cascade is prevented via formation of CD44-ezrin-actin-CD95 signaling complexes (*Mielgo et al., 2006*, *2007*). Likewise, in B cells the CD95-FADD interaction is prevented by binding of Btk to CD95 via its kinase and Pleckstrin homology (PH) domain (*Uckun, 1998*; *Vassilev et al., 1999*). Taken together, we reasoned that upon stimulation with CD95L, CD95 might assemble a signaling complex to induce integrin activation for myeloid cell rolling and adhesion.

Here, we report that CD95 mediates slow rolling and adhesion of myeloid cells via activation of integrin through stimulation of Syk-Btk-PLCγ2 or Btk-PLCγ2 signaling pathways. CD95 in myeloid cells or CD95L in endothelial cells is required for myeloid cell recruitment in in vivo animal models of inflammation. Further, deletion of CD95 in myeloid cells impairs bacterial clearance in systemic inflammation. Collectively, our data demonstrate that endothelial cell-derived CD95L serves as a chemokine in induction of neutrophil slow rolling and adhesion via integrin activation during inflammation.

## Results

### CD95L stimulation induces neutrophil slow rolling

In order to study the role of CD95 in leukocyte slow rolling we used a mouse autoperfused flow chamber assay (*Chesnutt et al., 2006*). This assay has the advantage of allowing examination of rolling and adhesion of neutrophils from whole blood, thereby preventing isolation-induced activation (*Forsyth and Levinsky, 1990*; *Glasser and Fiederlein, 1990*). In addition, using the *Lyz2*[CreGFP] reporter mice, 89 ± 2% of the rolling cells in the flow chamber have been identified as neutrophils (*Chesnutt et al., 2006*). Consistent with previous reports, the rolling velocity of neutrophils is

significantly reduced on E-selectin+ICAM1-coated chamber as compared to the E-selectin-coated chamber (*Figure 1A*, *Figure 1—figure supplement 1A*) (*Jung and Ley, 1999*; *Chesnutt et al., 2006*; *Zarbock et al., 2007*). Additional intravenous tail (i.v.) injection of CD95L one hour prior to flow chamber assay significantly reduced the rolling velocity as compared to control counterparts (*Figure 1B,C*; $1.55 \pm 0.07$ μm/s vs. $1.16 \pm 0.03$ μm/s). In order to exclude the possibility that CD95 mediates neutrophil slow rolling via the CD95-induced chemokine production, which has been reported in various cell types (*Park et al., 2003*; *Guo et al., 2005*; *Altemeier et al., 2007*; *Miwa et al., 1998*), mouse blood was perfused through the flow chambers coated with E-selectin, ICAM1 and CD95L. The rolling velocity of neutrophils in 50 μg/ml CD95L-coated chamber was significantly lower than in the control group (*Figure 1B,C*; $1.55 \pm 0.07$ μm/s vs. $0.84 \pm 0.09$ μm/s). Flow chamber coated with 50 μg/ml CD95L showed the strongest effect on slow rolling as compared to other coating concentrations (*Figure 1—figure supplement 1B*). Rolling cells in CD95L alone-coated or CD95L/ICAM1-coated chamber were not detectable which indicated that CD95L-induced slow rolling was E-selectin-dependent (data not shown). In addition, CD95L i.v. injection or CD95L-coating increased the number of rolling cells in the flow chamber as compared to the control group (*Figure 1D*). To further confirm that the CD95L-induced neutrophil slow rolling was specific to CD95, we specifically deleted CD95 in myeloid cells ($Fas^{<f/f>}::Lyz2^{<Cre>}$). Although CD95-deficient neutrophils rolled at a similar velocity as $Fas^{<f/f>}$neutrophils (*Figure 1—figure supplement 1A*), CD95L i.v.-injection or CD95L coating failed to reduce neutrophils' rolling velocity in $Fas^{<f/f>}::Lyz2^{<Cre>}$ mice (*Figure 1B*). Interestingly, $Fas^{<f/f>}::Lyz2^{<Cre>}$ mice showed significantly less rolling cells in CD95L-coated flow chamber or upon CD95L injection as compared to the *WT* mice under the same condition (*Figure 1D*). Control experiments demonstrated that $Fas^{<f/f>}::Lyz2^{<Cre>}$ mice exhibited less rolling cells in a flow chamber coated with E-selectin and ICAM1 than $Fas^{<f/f>}$mice, however this was not significant (*Figure 1—figure supplement 1C*). These results imply that CD95 might be important for the arrest of neutrophils.

More importantly, the effect of coated CD95L on neutrophil slow rolling was blocked by an integrin $\alpha_L$ neutralizing antibody, anti-CD11a, indicating that CD95L-induced slow rolling was integrin $\alpha_L$-dependent (*Figure 1E*). However, integrin $\alpha_M$ neutralizing antibody, anti-CD11b, did not block CD95L-induced slow rolling (*Figure 1—figure supplement 1D*).

In order to examine whether CD95 is also involved in L- and P-selectin-mediated rolling, we performed the autoperfused flow chamber assay with chambers coated with L/P-selectin, ICAM1 and CD95L respectively. CD95L stimulation did not significantly impact the rolling velocity in L-selectin or P-selectin coated chambers (*Figure 1—figure supplement 1E,F*).

To further evaluate the effect of CD95-induced rolling and adhesion in vivo, we conducted intravital microscopy of the inflamed cremaster muscle from $Fas^{<f/f>}$ or $Fas^{<f/f>}::Lyz2^{<Cre>}$ mice 2 hr after administration of tumor necrosis factor $\alpha$ (TNF-$\alpha$; 500 ng/mice intrascrotally, *Figure 1F*). It has been reported that >95% of all adherent and rolling leukocytes are neutrophils in this model (*Jung et al., 1998*). Interestingly, the rolling velocity of leukocyte in $Fas^{<f/f>}::Lyz2^{<Cre>}$ mice was not reduced (*Figure 1G*), indicating a redundant function of TNF-$\alpha$ and CD95 in modulation of rolling velocity, similar to the redundancy previously reported in a model of traumatic brain injury in mice (*Bermpohl et al., 2007*). Importantly, two studies showed that TNF was involved in neutrophil and T-cell adhesion via TNF-induced inside-out signaling (*Lauterbach et al., 2008*; *Li et al., 2016*). In order to clarify the redundant effect of TNF on CD95-deficiency, we stained neutrophils from blood of $Fas^{<f/f>}$ and $Fas^{<f/f>}::Lyz2^{<Cre>}$ mice for TNF receptors (TNFR) and observed that naïve $Fas^{<f/f>}::Lyz2^{<Cre>}$ mice expressed higher levels of TNFR2 but similar levels of TNFR1. However, at 6 hr after cecal ligation and puncture (CLP), neutrophils from $Fas^{<f/f>}::Lyz2^{<Cre>}$ mice had higher expression of TNFR1 but similar expression of TNFR2 (*Figure 1—figure supplement 2*). Thus, increased TNFR1 expression upon inflammation might compensate for the lack of CD95.

Consistent with our previous report that CD95 triggers transmigration of myeloid cells to the inflammatory site (*Letellier et al., 2010*), the numbers of adherent cells and transmigrated cells in $Fas^{<f/f>}::Lyz2^{<Cre>}$ mice were reduced as compared to $Fas^{<f/f>}$ mice (*Figure 1H, and I*),

Neutrophil slow rolling is mainly mediated by activation of lymphocyte function-associated antigen 1 (LFA-1, Integrin $\alpha_L\beta_2$) (*Chesnutt et al., 2006*; *Zarbock et al., 2007*). Therefore, integrin $\alpha_L$, $\alpha_M$ and $\beta_2$ surface expression levels on neutrophils were assessed by flow cytometry in whole blood of CD95L-injected and control mice. No significant difference between these groups could be detected (*Figure 1—figure supplement 3A–D*). Similarly, in $Fas^{<f/f>}::Lyz2^{<Cre>}$ mice (*Figure 1—figure*

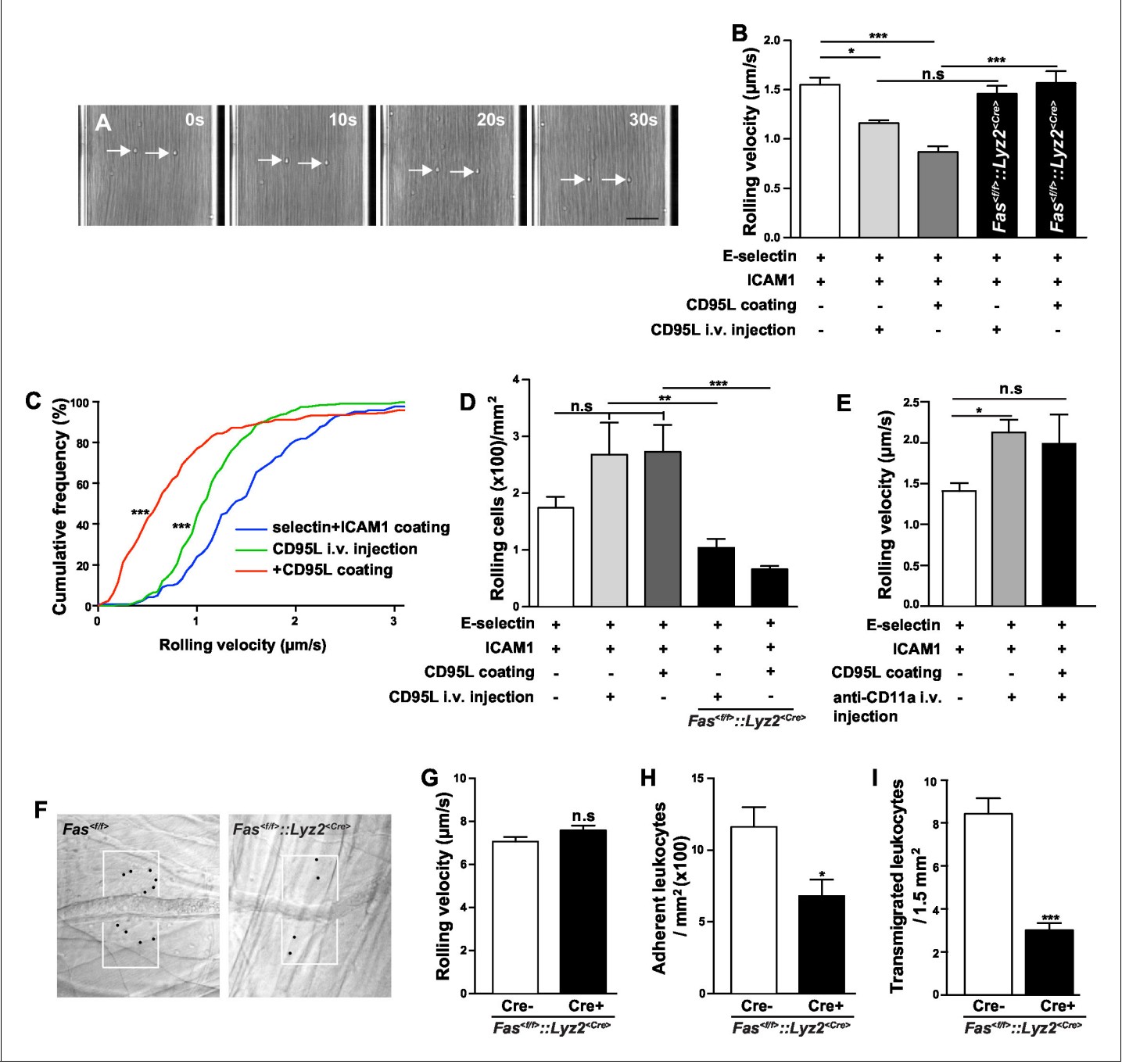

**Figure 1.** CD95 signaling in myeloid cells is involved in mediating slow rolling, adhesion and transmigration. (**A**) Representative time lapse pictures of neutrophil slow rolling in flow chamber. Arrows indicate the rolling cells. Scale bar: 50 µm. (**B**) Rolling velocity of *WT* or *Fas<sup><f/f></sup>::Lyz2<sup><Cre></sup>* neutrophils in flow chambers upon the stimulation of immobilized CD95L or soluble CD95L. Data are presented as mean ± SEM, n=3–4. (**C**) Cumulative histogram shows the velocity of rolling neutrophils in flow chambers coated with E-selectin/ICAM1, E-selectin/ICAM1/CD95L or E-selectin/ICAM1+soluble CD95L stimulation. (**D**) Number of *WT* or *Fas<sup><f/f></sup>::Lyz2<sup><Cre></sup>* rolling cells in flow chambers upon the stimulation of immobilized CD95L or soluble CD95L. Data are presented as mean ± SEM, n=3–4. (**E**) Rolling velocity of neutrophils in flow chambers coated with E-selectin/ICAM1 in the presence of immobilized CD95L or anti-CD11a antibody. Data are presented as mean ± SEM, n=3. (**F**) Representative reflected light oblique transillumination pictures of postcapillary venules of *Fas<sup><f/f></sup>* and *Fas<sup><f/f></sup>::Lyz2<sup><Cre></sup>* mice 2 hr after TNF-α application. Demarcations on each side of the venule determine the areas in which extravasated leukocytes were counted. (**G–I**) Rolling velocity of leukocytes (**G**) and numbers of adherent leukocytes (**H**) in the inflamed cremaster muscle venules and numbers of transmigrated leukocytes (**I**) in inflamed cremaster muscle of *Fas<sup><f/f></sup>* and *Fas<sup><f/f></sup>::Lyz2<sup><Cre></sup>* mice. Data are presented as mean ± SEM, n=6. Statistical significance was evaluated by one-way ANOVA followed by Bonferroni multiple comparison post hoc test in

*Figure 1 continued on next page*

Figure 1 continued

(B, C, D, E) (F=13.44, p<0.0001 in B, F=37.37, p<0.0001 in C, F=10.21, p<0.0001 in D, F=4.40, p=0.0135 in E) and two-tailed unpaired Student's *t* test in (G–I), *p<0.05, **p<0.01, ***p<0.001, n.s not significant.

The following figure supplements are available for figure 1:

**Figure supplement 1.** Rolling velocity of *WT* or *Fas$^{<f/f>}$::Lyz2$^{<Cre>}$* neutrophils in different conditions.

**Figure supplement 2.** TNFRs surface expression level of neutrophils from *Fas$^{<f/f>}$* and *Fas$^{<f/f>}$::Lyz2$^{<Cre>}$* mice in homeostasis and inflamed conditions.

**Figure supplement 3.** CD95L i.v. injection or deletion of CD95 in myeloid cells doesn't influence the integrin level in neutrophils.

*supplement 3E*), there was no difference in integrin $\alpha_L$ and $\beta_2$ expression between CD95-deficient neutrophils and *Fas$^{<f/f>}$* neutrophils, but only an increase of integrin $\alpha_M$ levels in CD95-deficient neutrophils (*Figure 1—figure supplement 3F–I*). CD95-deficient neutrophils also expressed the same level of CD44, the ligand of E-seletin and P-selectin, as the *Fas$^{<f/f>}$* neutrophils (*Figure 1—figure supplement 3J*). In addition, the ratio of neutrophils and monocytes increased in the blood of *Fas$^{<f/f>}$::Lyz2$^{<Cre>}$* mice, but T cells and B cells were not changed as compared to the control mice, and the absolute number of neutrophils in the blood of *Fas$^{<f/f>}$::Lyz2$^{<Cre>}$* mice was similar to the control counterparts (*Figure 1—figure supplement 3K,L,M*). These results show that CD95-induced slow rolling is not related to the up-regulation of cell surface expression level of integrins.

## Endothelial cells-derived CD95L mediates neutrophil recruitment

In inflamed tissue, inflammatory cytokines activate the expression of adhesion molecules, such as selectin and ICAM, and the synthesis of chemokines and lipid chemoattractants on the luminal surface of endothelial cells to facilitate the recruitment of leukocytes (*Ley et al., 2007*). In this study, we show that in the autoperfused flow chamber assay immobilized CD95L promotes the slow rolling of neutrophils in an integrin signaling-dependent pathway. Hence, we hypothesized that in vivo activated endothelial cells present CD95L to facilitate neutrophil recruitment. In order to address this hypothesis, we crossed *Cdh5$^{<CreERT2>}$* mice with *Fasl$^{<f/f>}$* mice to enable inducible deletion of CD95L in endothelial cells by tamoxifen treatment (*Fasl$^{<f/f>}$::Cdh5$^{<CreERT2>}$*) and performed the intravital microscopy experiments as we did with *Fas$^{<f/f>}$::Lyz2$^{<Cre>}$* mice (*Figure 2A,B*, *Figure 2—figure supplement 1A,B*). Interestingly, the leukocyte rolling velocity was significantly increased in mice with CD95L deficiency in endothelial cells as compared to the *Fasl$^{<f/f>}$* mice (*Figure 2C*). Moreover, the rolling flux fraction which shows the percentage of rolling cells was reduced in *Fasl$^{<f/f>}$::Cdh5$^{<CreERT2>}$* mice (*Figure 2D*). These observations indicate that endothelial cell-derived CD95L is essential for leukocyte slow rolling during inflammation. In line with the results from *Fas$^{<f/f>}$::Lyz2$^{<Cre>}$* mice, the numbers of adherent cells in the inflamed venules and transmigrated cells in the cremaster muscle were also reduced in *Fasl$^{<f/f>}$::Cdh5$^{<CreERT2>}$* mice as compared to the control litter mates (*Figure 2E, F*).

To further prove that endothelial cell-derived CD95L is required for myeloid cell recruitment, we used a thioglycollate-induced peritonitis model. Neutrophil extravasation into the inflamed peritoneal cavity was assessed in endothelial-CD95L deficient mice (*Figure 2G,H*). The number of peritoneal neutrophils at 6 hr after thioglycollate injection was significantly reduced in endothelial-CD95L deficient mice as compared to control counterparts (*Figure 2I*). Nonetheless, deletion of CD95 in the endothelial compartment (*Fas$^{<f/f>}$::Cdh5$^{<CreERT2>}$*) did not affect thioglycolate-induced neutrophil recruitment to the inflamed peritoneum (*Figure 2J*, *Figure 2—figure supplement 1C*). Notably, the cell surface expression of adhesion molecules, E-selectin, ICAM1 and ICAM2 was similar in CD95L-deleted and non-deleted endothelial cells, with the exception of cell surface expression of P-selectin that was reduced (*Figure 2—figure supplement 1D–G*). Similarly, deletion of CD95 in myeloid cells had no impact on ICAM1 and selectin levels in endothelial cells (*Figure 2—figure supplement 1H–J*). Additionally, the ratio of different leukocyte populations in the blood and the absolute number of blood neutrophils were not significantly changed in *Fasl$^{<f/f>}$::Cdh5$^{<CreERT2>}$* mice before or after tamoxifen induction as compared to the control mice (*Figure 2—figure supplement 2A,B,C*). These data demonstrate that the involvement of endothelial cell-derived CD95L in

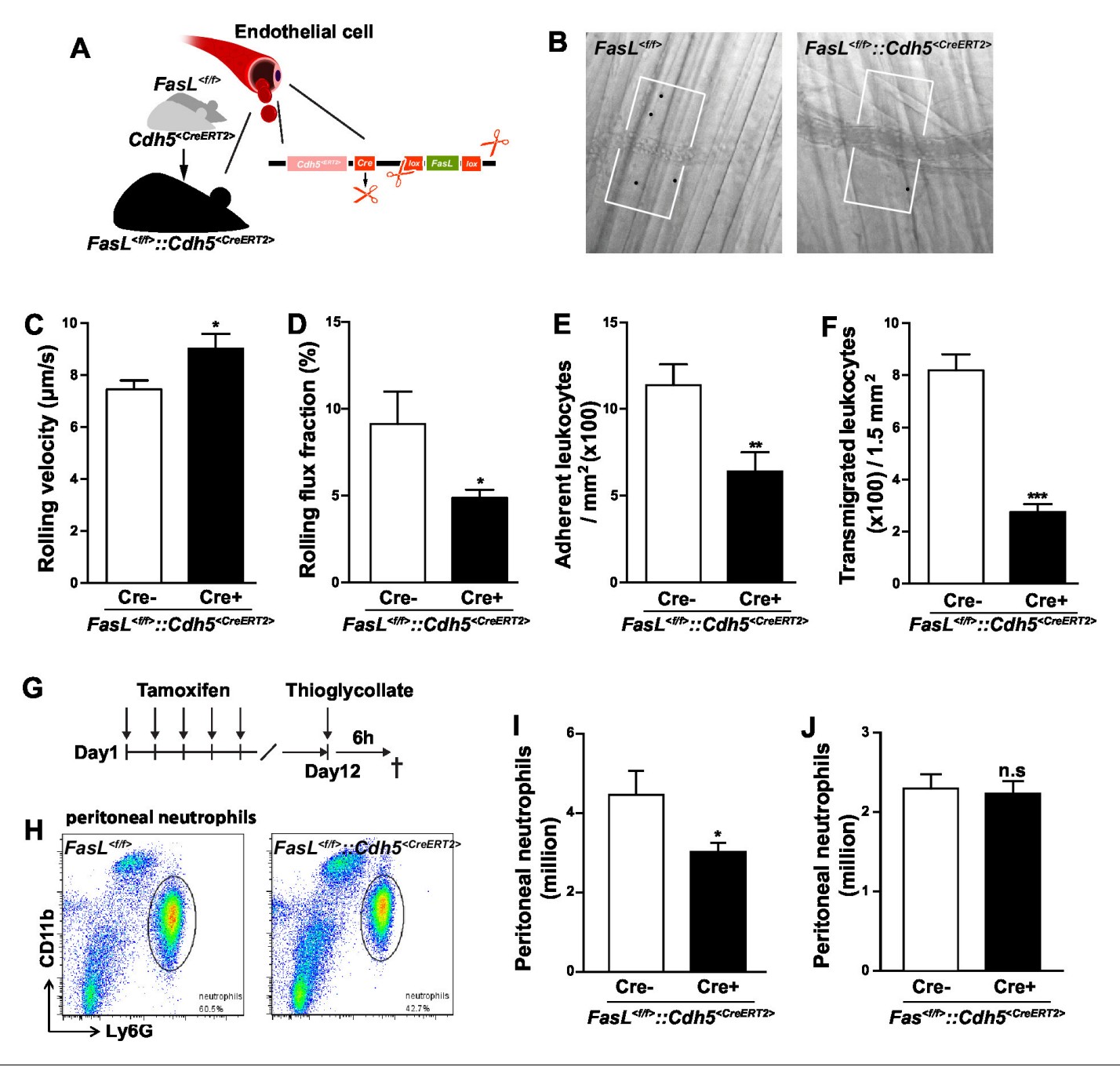

**Figure 2.** Endothelial cells-derived CD95L is necessary for neutrophil recruitment during inflammation. (**A**) Scheme of inducible CD95L deletion in endothelial cells of $Fasl^{<f/f>}::Cdh5^{<CreERT2>}$ mouse line. (**B**) Representative reflected light oblique transillumination pictures of postcapillary venules of $Fasl^{<f/f>}$ and $Fasl^{<f/f>}::Cdh5^{<CreERT2>}$ mice 2 hr after TNF-α application. Demarcations on each side of the venule determine the areas in which extravasated leukocytes were counted. (**C–F**) Rolling velocity of leukocytes (**C**), rolling flux fraction (**D**) and numbers of adherent leukocytes (**E**) in inflamed cremaster muscle venules and numbers of transmigrated leukocytes in inflamed cremaster muscle (**F**) of $Fasl^{<f/f>}$ and $Fasl^{<f/f>}::Cdh5^{<CreERT2>}$ mice. Data are presented as mean ± SEM, n=6. (**G**) Injection schedule of tamoxifen and thioglycollate is depicted. (**H**) Flow cytometry plot of peritoneal neutrophils at 6 hr after thioglycollate injection. (**I**) Influx of peritoneal neutrophils 6 hr after thioglycollate injection in $Fasl^{<f/f>}$ and $Fasl^{<f/f>}::Cdh5^{<CreERT2>}$ mice. n=11–14. (**J**) Influx of peritoneal neutrophils 6 hr after thioglycollate injection in $Fasl^{<f/f>}$ and $Fasl^{<f/f>}::Cdh5^{<CreERT2>}$ mice. Data in I–J are presented as mean ± SEM and were pooled from two independent experiments, n=16–17. Statistical significance was evaluated by two-tailed unpaired Student's t test in (**C–F**, **I**, **J**), *p<0.05, **p<0.01, ***p<0.001, n.s not significant.

The following figure supplements are available for figure 2:

*Figure 2 continued on next page*

*Figure 2 continued*

**Figure supplement 1.** Induced-deletion of CD95L or CD95 has no influence on ICAM and E-selectin level in endothelial cells.

**Figure supplement 2.** Characterization of *Fasl<f/f>::Cdh5<CreERT2>* mice.

leukocyte slow rolling and transmigration is not related to the change of adhesion molecule expression level on the luminal surface of blood vessels or the homeostasis of leukocytes in the blood stream.

## Phosphorylation of Syk, Btk and PLCγ2 upon CD95L engagement

E-selectin engagement triggers a signaling cascade which cooperates with chemokine signals to facilitate neutrophil rolling and adhesion during inflammation (*Zarbock et al., 2007*). E-selectin is expressed by inflamed endothelial cells and engages PSGL-1, CD44 and other ligands in neutrophils (*Xia et al., 2002*; *Katayama et al., 2005*). Upon ligand engagement, it has been reported that E-selectin activates SFKs which in turn initiate a signaling pathway involving the activation of ITAM bearing adaptors, Syk, Btk, PLCγ2, P38 and PI3Kγ (*Yago et al., 2010*; *Mueller et al., 2010*). We have previously shown that CD95L triggers the recruitment of myeloid cells to inflammatory sites via SFK-Syk-PI3K pathway (*Letellier et al., 2010*). To validate whether CD95 signaling can also activate Btk and PLCγ2 via Syk, we studied Btk and PLCγ2 activation upon CD95L stimulation. As *Syk* deficiency (*Syk-/-*) is perinatal-lethal in mice (*Turner et al., 1995*), we cultured primary embryonic liver-derived macrophages. Activation of CD95 increased phosphorylation of Syk, Btk and PLCγ2 (*Figure 3A* and quantified analysis in *Figure 3B,C*). CD95-mediated phosphorylation of PLCγ2 greatly decreased in macrophages isolated from *Syk-/-* mice as compared to wild type (*WT*) cells (*Figure 3A,C*). The reduced basal level of p-PLCγ2 in *Syk-/-* cells indicates that Syk is not only involved in CD95L-mediated phosphorylation of PLCγ2 but also acts as a hub for other ligands. However, the upregulated phosphorylation of Btk was still present in *Syk-/-* as compared to *WT* cells (*Figure 3A,B*). Activation of Btk could be explained by the previously observed interaction of the PH domain of Btk with CD95 in B-cells (*Vassilev et al., 1999*). To examine this interaction we pulled down Btk from the lysates of CD95L-treated macrophages by immunoprecipitation. Binding of CD95 to Btk was detected following CD95L stimulation (*Figure 3D*). In order to validate the functional involvement of Btk in CD95L-induced PLCγ2 phosphorylation, we used the Btk inhibitor PCI-32765 (Ibrutinib) 1 hr prior to CD95L stimulation of macrophages. PCI-32765 fully blocked the basal and CD95-induced phosphorylation of Btk and PLCγ2 (*Figure 3E,F,G*), but not the phosphorylation of Syk (*Figure 3H*), which indicates that the phosphorylation of PLCγ2 is Btk-dependent and that Btk is an essential mediator for CD95-induced PLCγ2 activation. Taken together, these data demonstrate the presence of two signaling branches downstream of CD95: CD95/SFK/Syk/Btk/PLCγ2 and CD95/SFK/Btk/PLCγ2 (*Figure 3I*). More importantly, immobilized CD95L-induced slow rolling was abolished in PCI-32765 pretreated mice in the autoperfused flow chamber assay, implying that CD95L-induced slow rolling was Btk-dependent (*Figure 3J*).

## CD95L stimulation activates integrin

The common final step for integrin activation has been revealed as binding of talin1 and kindlin-3 to the cytoplasmic domain of β integrin which in turn breaks the salt bridges between the cytosolic domains of integrin α and β subunits and induces integrin conformational changes (*Tadokoro et al., 2003*; *Wegener et al., 2007*; *Lefort et al., 2012*). Recruitment of talin1 to LFA-1 is Rap1a-dependent (*Lefort et al., 2012*). In order to find out whether CD95 also activates Rap1, we performed an active Rap1 pull-down assay in CD95L-stimulated mouse bone marrow-derived neutrophils. Significant activation of Rap1 was observed in neutrophils 15 min after CD95L stimulation (*Figure 4A,B*). Of note, CD95L-treated and control neutrophils exhibited similar levels of integrin $\alpha_L$, integrin $\alpha_M$ and integrin $\beta_2$ expression (*Figure 4—figure supplement A–D*).

Inside-out integrin signaling triggers conformational changes in integrin, leading to increased binding affinity (integrin activation) and avidity (*Abram and Lowell, 2009*). Different conformations of LFA-1 can be recognized by integrin epitope specific antibodies. KIM127 recognizes an epitope

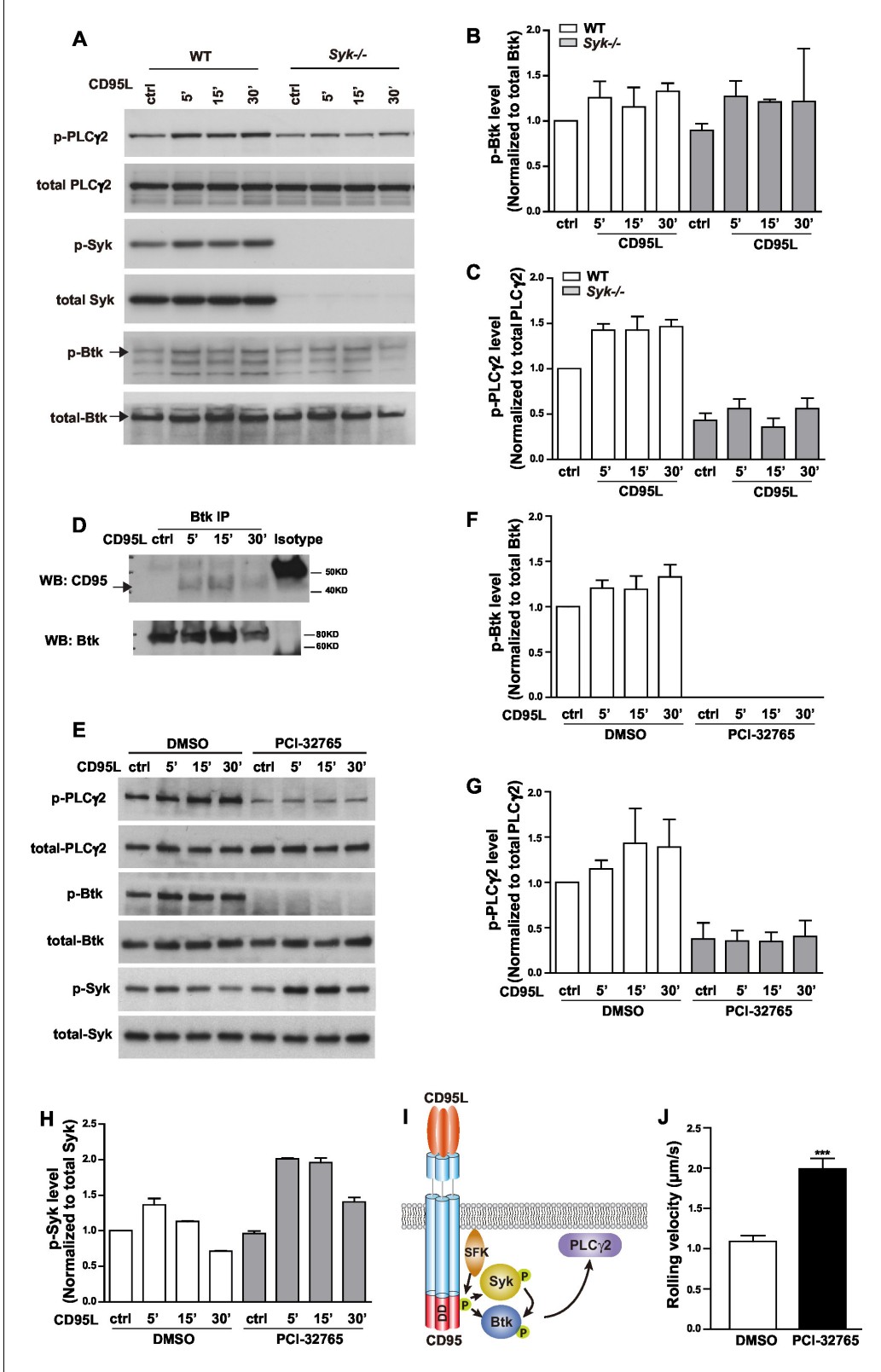

**Figure 3.** CD95L stimulation induces phosphorylation of PLCγ2 via activating Syk and Btk in myeloid cells. (**A**) Macrophages cultured from *WT* or *Syk-/-* embryonic liver cells were treated with CD95 (40ng/ml). Lysates were prepared at the indicated time points and immunoblotted for the indicated proteins. (**B–C**) Quantification analysis of PLCγ2 and Btk phosphorylation level in (**A**) from three independent experiments. Data are presented as mean ± SEM, n=3. (**D**) Macrophages cultured from bone marrow cells were treated with CD95L (40 ng/ml). Lysates were prepared at the indicated time and

*Figure 3 continued on next page*

*Figure 3 continued*

immunoprecipitated with anti-Btk followed by immunoblotting with CD95 and Btk antibody. (E) Macrophages cultured from bone marrow cells were treated with DMSO or Btk inhibitor PCI-32765 (1 μM) one hour prior to CD95L stimulation (40 ng/ml). Lysates were prepared at the indicated time points and immunoblotted for the indicated proteins. (F–H) Quantification analysis of Btk, PLCγ2 and Syk phosphorylation level in (E) from three independent experiments. Data are presented as mean ± SEM, n=3. (I) Scheme of CD95L stimulation-induced PLCγ2 activation. (J) Rolling velocity of neutrophils from DMSO or Btk inhibitor pre-treated mice in a flow chamber coated with E-selectin/ICAM1/CD95L. Data are presented as mean ± SEM, two-tailed unpaired Student's $t$ test, ***p<0.001, n=3.

of $\beta_2$ subunit of human LFA-1 when it is extended (*Beglova et al., 2002*), whereas mab24 binds to the epitope of I-like domain in $\beta_2$ subunit of human LFA-1 only if this is accessible as in the high-affinity state (*Lu et al., 2001*). To examine if CD95 induces integrin conformational changes, binding of reporter antibodies was analyzed by flow cytometry in U937 cells. To this end, cells were pre-incubated with the reporter antibodies, and thereafter perfused through the flow chamber in the presence of soluble or immobilized CD95L. We observed significantly increased binding of KIM127 and mab24 in cells treated with soluble CD95L (*Figure 4C,D*, *Figure 4—figure supplement 1E*). Notably, in the absence of coated E-selectin, soluble CD95L showed no effect on KIM127 and mab24 binding (*Figure 4—figure supplement 1F–G*). These data indicate that CD95L treatment induces integrin conformational changes including its extension and full activation in an E-selectin-dependent manner.

The soluble ICAM1 binding assay is a commonly used test for LFA-1 function, which is determined by affinity and avidity (*Salas et al., 2004*; *Lefort et al., 2012*). To further confirm that CD95L stimulation induces integrin activation, mouse bone marrow-derived neutrophils were incubated with ICAM1-Fc and the binding of ICAM1 was assessed by flow cytometry. Anti-CD11b antibody was used to block the integrin $\alpha_M\beta_2$ (Mac-1)-dependent ICAM1 binding. CD95L-activated neutrophils showed significant binding of soluble ICAM1 as compared to the non-treated group (*Figure 4E*). Taken together, these results demonstrate that CD95 signaling induces integrin activation.

## CD95 recruits and associates with integrin

Compartmentalization of multi-molecular signaling complexes integrates extracellular signals and facilitates integrin activation (*Bezman and Koretzky, 2007*). Specifically, a signalosome consisting of Src family kinase Hck, Btk, WASp and PLCγ2 has been identified to be indispensable for fMLF-induced MAC-1 activation required for neutrophil recruitment (*Volmering et al., 2016*). To test if CD95 assembles a signaling complex with LFA-1 that together with E-selectin orchestrates integrin activation, we performed proximity ligation assay of integrin $\alpha_L$ and CD95 in control or CD95L-treated dHL60 cells. Proximity ligation assay (PLA) detects interaction of proteins that are 30nm or less apart from each other (*Söderberg et al., 2006*). Indeed, binding of CD95 to integrin $\alpha_L$ was detected by PLA in dHL60 cells (*Figure 4F*). CD95L treatment significantly enhanced this binding, as shown by increased number of PLA signal of CD95-integrin clusters (*Figure 4G*) and a ratio of PLA signal positive cells (*Figure 4H*). Among the PLA positive cells, CD95L treatment increased the number and integrated density of PLA signal as compared to the non-treated control (*Figure 4I,J*). Interestingly, CD95L-treated cells also showed a tendency towards increased polymerization of F-actin as shown by phalloidin staining compared to the control cells (*Figure 4—figure supplement 1H*). Moreover, immunoprecipitation of integrin $\alpha_L$ from lysates of CD95L-treated mouse macrophages, revealed a stimulation-dependent association of CD95 with integrin $\alpha_L$ (*Figure 4K*). However, integrin $\alpha_M$ did not bind to CD95 upon CD95L stimulation (*Figure 4—figure supplement 1I*).

Collectively, our data indicate that activation of CD95 leads to the assembly of multiprotein complexes including integrins and that CD95 mediates integrin activation required for rolling and adhesion of myeloid cells.

## CD95 in myeloid cells is required for bacterial clearance

In order to test the involvement of myeloid cell-specific CD95 in systemic inflammation, we used a CLP-induced animal model of sepsis. Bacterial load in blood, spleen and peritoneal cavity of $Fas^{<f/f>}$ and $Fas^{<f/f>}::Lyz2^{<Cre>}$ mice was assessed by the number of colony-forming unit (CFU) 6 hr after CLP. Importantly, $Fas^{<f/f>}::Lyz2^{<Cre>}$ mice demonstrated significantly higher CFU in peritoneal lavage

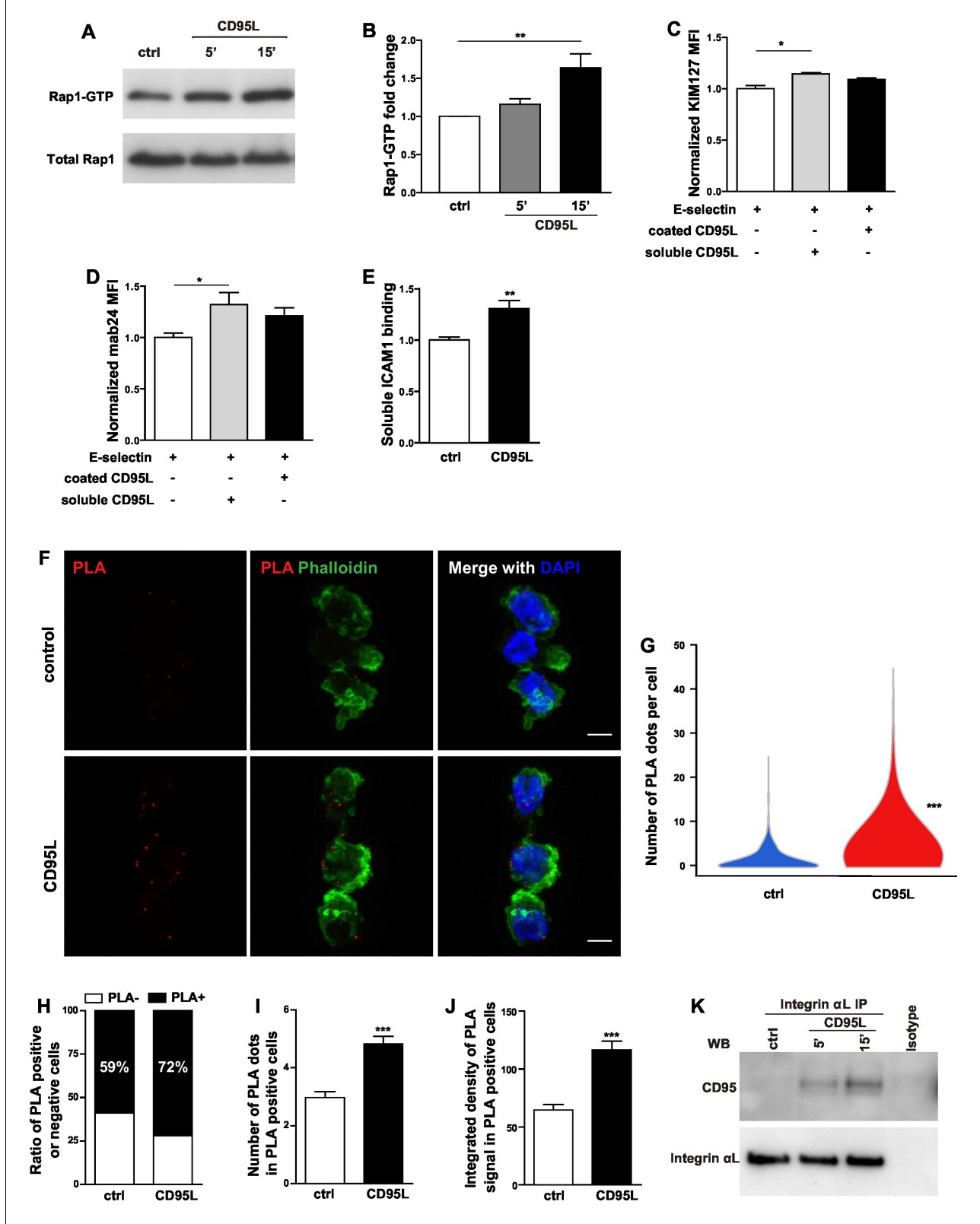

**Figure 4.** CD95L stimulation induces integrin activation and recruitment of integrin to CD95. (**A**) Bone marrow-derived murine neutrophils were treated with CD95L (40 ng/ml). Lysates were prepared at the indicated time points and GST-RalGDS-RBD peptide affinity-precipitated for Rap1 immunoblotting. (**B**) Quantification analysis of Rap1-GTP activation in (**A**) from three independent experiments. Data are presented as mean ± SEM, n=3. (**C–D**) U937 cells were perfused through human E-selectin coated flow chamber in the presence of soluble or immobilized CD95L. The binding of

*Figure 4 continued on next page*

*Figure 4 continued*

KIM127 (**C**) or mAb24 (**D**) were analyzed by flow cytometry and presented as mean ± SEM, n=3. (**E**) Bone marrow-derived murine neutrophils were treated with coated CD95L for 10 min. The binding of soluble ICAM1 was analyzed by flow cytometry and data presented as mean ± SEM, n=3. (**F**) Representative pictures show PLA of integrin $\alpha_L$ and CD95 in control or CD95L-treated dHL60 cells. Red, PLA; green, Phalloidin; blue, DAPI. Scale bar: 10 μm. (**G**) The number of PLA signal in each control or CD95L-treated dHL60 cell. Data are presented as violin plot, 383 control cells and 630 CD95L-treated dHL60 cells from 8 random fields were evaluated. (**H**) Ratio of PLA negative and positive cells in control or CD95L-treated dHL60. Data are presented as stacked bar. (**I–J**) Number of PLA signal (**I**) and integrated density (**J**) of PLA signal in PLA positive cells. Data are presented as mean ± SEM, n=226–453. (**K**) Bone marrow-derived macrophages were treated with CD95L (40 ng/ml). Lysates were prepared at the indicated time and immunoprecipitated with anti-CD11a followed by immunoblotting for CD95 and CD11a antibody. Statistical significance was evaluated by one-way ANOVA followed by Bonferroni multiple comparison post hoc test in (**B–D**) and two-tailed unpaired Student's $t$ test in (**E, G, I, J**), *$p<0.05$, **$p<0.01$, ***$p<0.001$, n.s not significant.

The following figure supplement is available for figure 4:

**Figure supplement 1.** CD95L treatment doesn't influence the integrin level of neutrophils in vitro.

liquid and blood as compared to the $Fas^{<f/f>}$ littermate control mice (**Figure 5A,B**). CFU in spleen homogenate was also elevated, but was not significantly higher (**Figure 5C**). Moreover, peritoneal neutrophil infiltration in $Fas^{<f/f>}::Lyz2^{<Cre>}$ mice was reduced as compared to the $Fas^{<f/f>}$ littermate control mice (**Figure 5D**). These data indicate that CD95 in myeloid cells is involved in mounting an effective bacterial clearance response during systemic inflammation via recruiting neutrophils to the inflammatory sites.

In summary, inflammation induces CD95L expression in endothelial cells (**Sata and Walsh, 1998**). CD95 together with E-selectin orchestrate signaling events leading to integrin activation that finally result in slow rolling and adhesion of myeloid cells.

## Discussion

CD95-induced leukocyte infiltration was first found in early studies aiming at inducing apoptosis of tumor cells in vivo (**Arai et al., 1997**; **Seino et al., 1997**). In these studies, transplantation of CD95L-overexpressing/CD95-negative tumor cells induced a dramatic neutrophil infiltration into the tumor xenografts. Other studies using Boyden chamber assays demonstrated that soluble CD95L induces the transmigration of human neutrophils in vitro (**Seino et al., 1998**; **Ottonello et al., 1999**; **Dupont and Warrens, 2007**). Although these findings are interesting, they did not address whether CD95-induced neutrophil recruitment was through direct CD95 activation on neutrophils or secondary to CD95-induced production of inflammatory mediators. The present study shows that CD95 is involved in induction of slow rolling and adhesion of neutrophils, and that these steps are blocked in CD95-deficient neutrophils, indicating that CD95 induces slow rolling via a direct effect on neutrophils and not via induction of inflammatory cytokines and chemokines.

In this study, soluble or coated CD95L induce neutrophil slow rolling in the autoperfused flow chamber assay. However, coating of the flow chamber with CD95L alone was not sufficient to induce leukocyte tethering (capturing), indicating that E-selectin is required for CD95-mediated slow rolling. It has been shown that selectin ligands PSGL-1 and CD44 are enriched in lipid rafts (**Miner et al., 2008**; **Neame et al., 1995**). In addition, the three SFKs of neutrophils, Fgr, Hck, Lyn, which are activated upon the engagement of selectin to its ligands (**Yago et al., 2010**), also associate with cholesterol-dependent membrane rafts (**Lowell, 2004**). Interestingly, neutrophil slow rolling has been reported to be dependent on intact lipid microdomains to signal slow rolling on E-selectin and P-selectin (**Yago et al., 2010**). The clustering of lipid microdomains is regulated by the actin cytoskeleton (**Chichili and Rodgers, 2007**). Ezrin/radixin/moesin (ERM) proteins, which link the cytoskeleton to integral membrane proteins via their FERM domains, associate with PSGL-1 and CD44 through their cytoplasmic domains (**Yonemura et al., 1998**; **Urzainqui et al., 2002**). Moreover, ligation of PSGL-1 to selectin recruits Syk to an atypical ITAM on ERM proteins bound to the cytoplasmic domain of PSGL-1 (**Urzainqui et al., 2002**). Thus, leukocyte rolling requires the formation of multiprotein complexes at the plasma membrane.

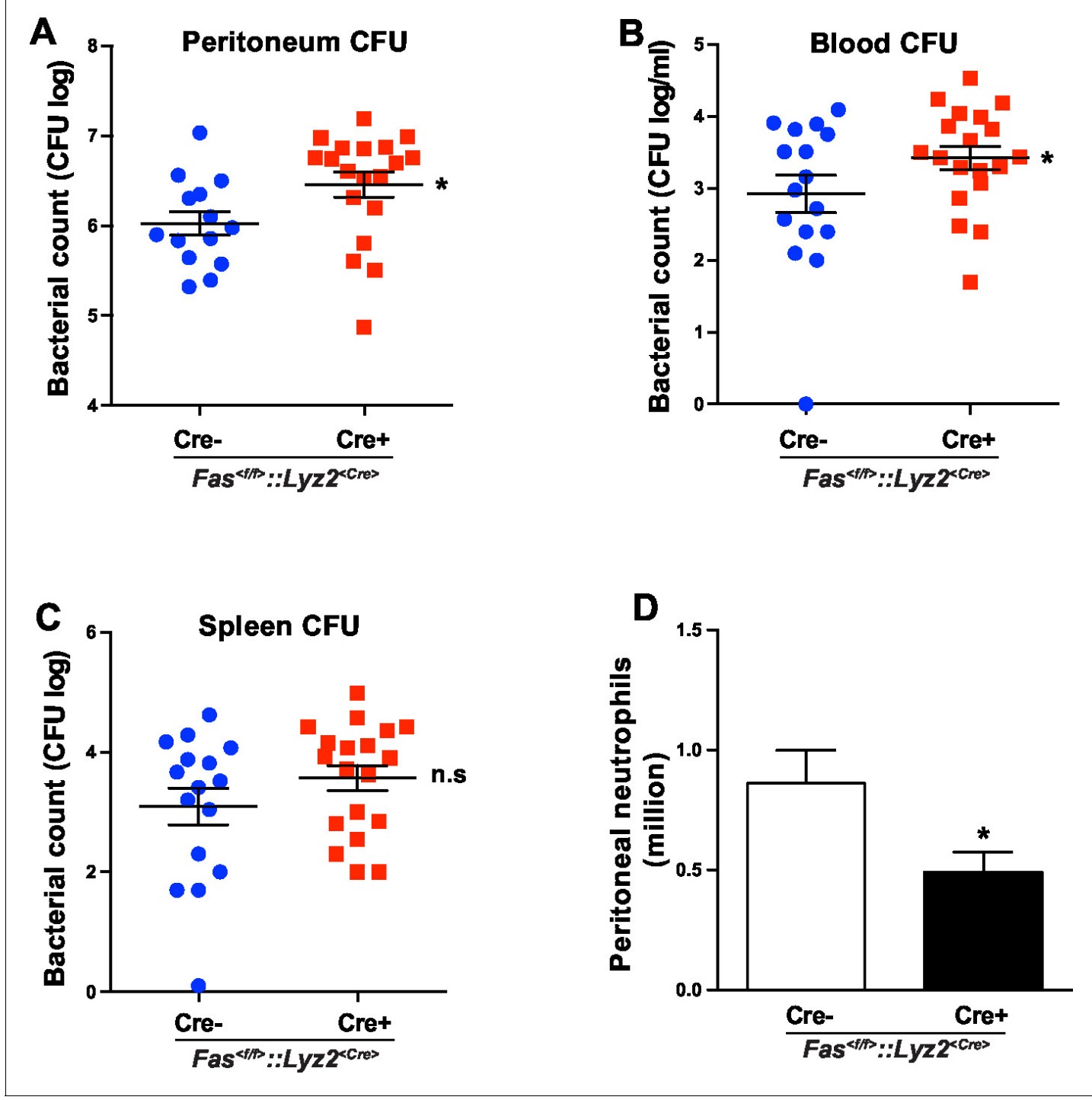

**Figure 5.** CD95 in myeloid cells is required for bacterial clearance. (**A–C**) Bacterial counts of peritoneal lavage fluid (**A**), blood (**B**) and spleen (**C**) from *Fas*<sup><f/f></sup> or *Fas*<sup><f/f></sup>*::Lyz2*<sup><Cre></sup> mice 6 hr after CLP. Data are pooled from three independent experiments and presented as dot plot with mean ± SEM, n=16–19. (**D**) Infiltrating peritoneal neutrophils 6 hr after CLP in *Fas*<sup><f/f></sup> or *Fas*<sup><f/f></sup>*::Lyz2*<sup><Cre></sup> mice. Data are pooled from three independent experiments and presented as mean ± SEM, n=14–18. Significance between groups was evaluated by running a linear mixed model for the log-CFU with the random covariable of time point and the fixed covariable of gender. *p<0.05, n.s not significant,

CD95 clustering is accompanied by reorganization of the actin cytoskeleton and aggregation of lipid microdomains (*Söderström et al., 2005*). Accordingly, CD95 clustering in sphingolipid-rich membrane microdomains is necessary for the induction of CD95 signaling (*Grassme et al., 2001*). CD95 indirectly associates with actin via direct and specific binding to ezrin FERM domains (*Lozupone et al., 2004*), and the organization of the microfilaments affects the outcome of CD95 stimulation (*Parlato et al., 2000*). In addition, CD44 has been reported to bind to CD95 via ezrin to block the apoptotic signal transduction (*Mielgo et al., 2006*, *2007*). These findings suggest that CD95 may also be a part of a multiprotein complex encompassing selectin ligands, SFKs and cytoskeletal proteins that orchestrate leukocyte rolling.

Ligand binding to external domains causes conformational changes that increase ligand affinity, and formation of integrin clustering, which in turn results in SFK autophosphorylation and Syk kinase activation in the outside-in integrin pathway (*Abram and Lowell, 2009*). SFK and Syk kinases can directly interact with the cytoplasmic domain of $\beta_2$, and $\beta_3$ integrins (*Arias-Salgado et al., 2003*). In addition, ITAM containing adaptor proteins DAP12 and FcR$\gamma$ couples Syk to integrins (*Mócsai et al., 2006*). The ITAM-like YXXL motif of CD95 is involved in the CD95-Lyn-Syk signaling cascade leading to myeloid cell transmigration (*Letellier et al., 2010*). Interestingly, the current study identifies an association of CD95 with integrin $\alpha_L\beta_2$ in neutrophils, as assessed by PLA, and macrophages, as confirmed by direct co-immunoprecipitation. Moreover, integrin activation reporter assay and the soluble ICAM1 binding assay demonstrate that CD95-induced integrin activation is a mechanism present in both macrophages and neutrophils. Altogether, these findings strongly indicate that CD95 modulates rolling and adhesion via its participation in a multiprotein signaling complex containing selectin ligands, SFKs, integrins and cytoskeletal proteins in both neutrophils and macrophages.

Alternatively, CD95 might indirectly impact myeloid cell recruitment by promoting secretion of inflammatory cytokines, as already reported in a variety of cell types (*Altemeier et al., 2007*; *Dupont and Warrens, 2007*; *Park et al., 2003*; *Wang et al., 2010a*). In line with this, a recent study showed that CD95 in apoptotic cells induced the production of pro-inflammatory cytokines and chemokines, which in turn attracted myeloid cells (*Cullen et al., 2013*). Interestingly, the autoperfused flow chamber assay in this study shows that CD95-induced slow rolling is independent of chemokine production.

The upregulation of selectins and ICAMs in endothelial cells and selectin- and ICAM-ligands in leukocytes play important roles in rolling and adhesion (*Pober and Sessa, 2007*). Unlike the effect of TNF-$\alpha$ on endothelial cells, CD95L stimulation or CD95/CD95L deletion in endothelial cells have no impact on the expression level of adhesion molecules in endothelial cells. A previous study has reported that crosslinking of CD95 with antibody rapidly triggers down-modulation of L-selectin, CD44, LFA$\alpha$ and LFA$\beta$ in CD95-sensitive T cell blasts (*Kabelitz et al., 1996*). In our study, in vitro or in vivo treatment with CD95L, or CD95 deletion in myeloid cells did not change cell surface expression levels of most integrins, which shows a cell type dependent effect of CD95 on the expression of adhesion molecules. These results demonstrate that CD95-induced neutrophil slow rolling is independent of regulation of adhesion molecules.

Endothelial cells play a pivotal role in leukocyte recruitment by synthesizing and presenting chemokines and leukocyte adhesion molecules during inflammation (*Pober and Sessa, 2007*). Interestingly, CD95L has also shown to be expressed by endothelial cells (*Sata and Walsh, 1998*). Overexpression of CD95L by adenovirus transduction in endothelial cells markedly attenuated TNF-$\alpha$-induced T cell and macrophage infiltration, and adherent mononuclear cells underwent apoptosis (*Sata and Walsh, 1998*). Along this line, tumor endothelial cells selectively and highly express CD95L, which serves as a barrier to prevent the infiltration of CD8 cells via induction of apoptosis in the establishment of immune tolerance (*Motz et al., 2014*). Contrary to the apoptotic effect of endothelial cell-derived CD95L, the present study shows that deletion of CD95L in endothelial cells impairs neutrophil recruitment in inflamed cremaster muscle and thioglycolate-induced peritonitis. Together, these data indicate that endothelial cell-derived CD95L may serve as a chemokine to induce myeloid cell recruitment during inflammation. Endothelial cells are known to basally express CD95L (*Sata and Walsh, 1998*), however, as shown by our study on CD95L-induced slow rolling in the autoperfused flow chamber assay, CD95L might only induce slow rolling in an inflammatory setting, as this function requires additional presentation of E-selectin by endothelial cells. In addition, inflammatory cytokines might increase CD95L levels in endothelial cells. Along this line, IFN$\gamma$ activates CD95L promoter activity in T-cells (*Kirchhoff et al., 2002*).

Integrin signaling plays important roles in regulating cancer 'stemness', metastasis and drug resistance (*Seguin et al., 2015*). As CD95 is now recognized as an inducer of tumor cell growth and invasion (*Martin-Villalba et al., 2013*; *Peter et al., 2015*), it is of great importance to study the CD95-induced integrin signaling in tumor progression. On the other hand, tumor growth and metastasis are promoted by myeloid-derived suppressor cells (MDSC) (*Condamine et al., 2015*). Blockade of endothelial cells-derived CD95L in order to inhibit MDSCs recruitment to tumor might be used as a potential strategy for cancer therapy.

Effective removal of infectious organisms is of utmost importance to attenuate the early onset of sepsis (*Bosmann and Ward, 2013*). It has been reported that mice with a global impairment of CD95 activity, $Fas^{lpr/lpr}$ and $Fas^{gld/gld}$ mice, develop severe diarrhoea and showed impaired bacterial clearance in a bacterial-induced gut infection model (*Pearson et al., 2013*). Yet, the use of a mouse with ubiquitous impairment of CD95 activity hindered clarification of the exact mechanism underlying this impairment. We now show that myeloid cells require CD95 activity to efficiently to infiltrate into the inflammatory sites and clear bacteria following CLP-induced sepsis. We observed increased bacterial load in blood and peritoneum of $Fas^{<f/f>}::Lyz2^{<Cre>}$ mice as compared to control counterparts, in a CLP-induced sepsis model. Altogether these data reveal that CD95 in myeloid cells plays an important role in bacterial clearance.

Altogether, this study shows a chemoattractant effect of endothelial cell-derived CD95L in induction of neutrophil slow rolling and adhesion via integrin activation. Both cancer cells and immune cells exhibit very high levels of CD95 surface expression. Therapies aimed at interfering with CD95's activity can be used for the treatment of diseases with a major cell-extravasation component such as cancer progression and inflammation.

# Materials and methods

## Animals

C57BL/6N mice were purchased from Charles River Laboratories. *Syk+/-* mice were from Martin Turner (The Babraham Institute, UK) and bred as heterozygous. $Fas^{<f/f>}$ mice (kind gift from K. Rajewsky, Max Delbrück Center for Molecular Medicine, Germany) were bred with $Lyz2^{<Cre>}$ (Jackson Laboratory, USA) mice. $Cdh5^{<CreERT2>}$ mice (Ralf H. Adams, University of Münster, Germany, *Wang et al., 2010b*) were bred with $Fasl^{<f/f>}$ or $Fas^{<f/f>}$ (*Karray et al., 2004*) mice. Animal experiments were performed in accordance with institutional guidelines of the German Cancer Research Center and were approved by the Regierungspräsidium Karlsruhe, Germany (Permit Number: G188/13).

## Autoperfused mouse flow chamber assay

Autoperfused mouse flow chambers assay was performed as previously reported (*Chesnutt et al., 2006*). Briefly, carotid artery of male, 12 weeks old, *WT* or $Fas^{<f/f>}::Lyz2^{<Cre>}$ mice was exposed and connected to flow chamber with a PE10 tubing. The free end of the flow chamber was connected to a water-filled PE50 tubing in order to control the pressure drop in the chamber which determined the shear stress of rolling cells. Flow chambers were coated with different combinations of 30 µg/ml E-selectin, 90 µg/ml L-selectin, 20 µg/ml P-selectin, 15 µg/ml ICAM1 (R&D systems, USA) and 50 µg/ml CD95L. For some conditions, mice were intravenously injected with 10 µg CD95L, 40 µg anti-CD11a antibody (M17/4, ebioscience, USA), 40 µg anti-CD11b antibody (M1/70, ebioscience, USA), DMSO (1:100) or Btk inhibitor (PCI-32765, Sigma, 15 mg/kg) in 100 µl saline one hour before performing the autoperfusion assay. Rolling cells in 3 random fields for each flow chamber were recorded and two flow chambers were used for each mouse. 3 to 4 mice were used for each group. The CD95L utilized in this study was a fusion protein of trimeric human CD95L-receptor binding domain fused with T4-Foldon motif from the fibritin of the bacteriophage T4 (CD95L-T4) and purified from CD95L-T4 plasmid-transfected HEK293T cells (*Kleber et al., 2008*). It is commercially available from IBA GmbH, Göttingen, Germany.

## Intravital microscopy

$Fas^{<f/f>}::Lyz2^{<Cre\pm>}$ and $Fasl^{<f/f>}::Cdh5^{<CreERT2\pm>}$ mice were used for intravital microscopy. 2 hr before cremaster muscle exteriorization, mice received 4 µg PTx i.v. (Sigma-Aldrich, USA) and 500

ng TNF-α intrascrotally (R&D systems, USA). Mice were anesthetized with an i.p. injection of 125 mg/kg ketamine hydrochloride (Sanofi), 0.025 mg/kg atropine sulfate (Fujisawa, Japan), and 12.5 mg/kg xylazine (Tranqui Ved; Phoenix Scientific, UK) and placed on a heating pad. The cremaster muscle was prepared as previously described (*Mueller et al., 2010*). Postcapillary venules with a diameter between 20 and 40 μm were recorded using an intravital microscope (Axioskop, SW 40/ 0.75 objective; Carl Zeiss, Inc.) through a digital camera. Blood flow centerline velocity was measured using a dual-photodiode sensor system (CircuSoft Instrumentation). Recorded images were analyzed using ImageJ and AxioVision (Carl Zeiss, Germany) software. Leukocyte rolling flux fraction was calculated as percentage of total leukocyte flux. Transmigrated cells were determined in an area reaching out 75 μm to each side of a vessel over a distance of 100 μm vessel length.

## Primary culture of mouse macrophages and neutrophils

Macrophages were cultured from femurs and tibias derived-bone marrow cells as previously described (*Letellier et al., 2010*) or from fetal liver cells of *Syk+/-* mice (E15). Neutrophils were isolated from mouse bone marrow cells over discontinuous 50%/55%/62%/81% percoll gradients. The gradients were centrifuged at 1600 g for 30 min without braking at 10°C and the interphase between 62% and 81% percoll was collected. Neutrophils were cultivated overnight in RPMI medium containing 10% fetal calf serum and 20% WEHI-3B conditioned medium.

## Thioglycollate-induced peritonitis

*Fasl*<sup>*<f/f>*</sup>::*Cdh5*<sup>*<CreERT2±>*</sup> and *Fas*<sup>*<f/f>*</sup>::*Cdh5*<sup>*<CreERT2±>*</sup> mice were gavaged with tamoxifen in sun flower seed oil (200 mg/kg, Sigma, Germany) for 5 consecutive days and 1ml of 3% thioglycollate broth (Fluka, Germany) in PBS was i.p. injected 7 days after the last tamoxifen induction. In this model, neutrophil infiltration peaks at 6 hr. Peritoneal cells were collected 6 hr after thioglycollate injection and total cells were counted as previously described (*Letellier et al., 2010*). Differential cell counts were accessed by flow cytometry after staining with neutrophil markers.

## Flow cytometry and cell type identification

Blood samples were stained with antibodies against leukocyte markers. Neutrophils were identified according to the profile of Forward Scatter (FSC)/Sider Scatter (SSC), DAPI-negativity, and CD45, CD11b, Ly6G-positivity. Neutrophils were stained with anti-CD95 (Jo2), anti-CD11b, anti-CD11a, anti-CD18 for testing the expression of CD95 and different integrins. Endothelial cells from dissociated liver cells were identified according to the profile of FSC/SSC, DAPI, CD45-negativity, and CD31-positivity. Antibodies of anti-CD95L (MFL3), anti-CD95, anti-ICAM1, anti-ICAM2, anti-E-selectin, and anti-P-selectin were used for testing the expression of CD95, CD95L and different adhesion molecules in endothelial cells. Flow cytometry data were analyzed with Flowjo software.

## Immunoprecipitation and western blotting

CD95L treated (40 ng/ml) or non-treated cells were washed with PBS containing phosphatase inhibitors, pelleted, and lysed on ice for 30 min with Pierce IP Lysis buffer (Fisher Scientific, Germany) containing vanadate, inhibitors for phosphatase and proteinase. Lysates of 500 μg protein were immunoprecipitated at 4°C for 4 hr with the anti-mouse CD11a (M17/4, biolegend), anti-mouse Btk (Cell Signaling, USA), anti-mouse CD11b (M1/70, ebioscience, USA) antibodies or the corresponding isotype controls. Afterward, 40 μl Dynabeads M-280 Streptavidin was added to each sample and incubated for 1 hr at 4°C with rotation. Beads were washed 5 times with 1 ml of lysis buffer. The immunoprecipitates were released by cooking the beads with 40 μl of 2x laemmli buffer at 95°C for 5 min.

Immunoblotting was performed as previously described (*Letellier et al., 2010*). Membranes were probed with following antibodies respectively: anti-phospho-Syk (Tyr319, 352), anti-phospho-Btk (Tyr223), anti-phospho-PLCγ2 (Tyr1217), anti-Syk, anti-Btk, anti-PLCγ2 (Cell Signaling, USA), anti-Rap1 (Fisher Scientific), anti-mouse CD11a, anti-CD95 (M20, Santa Cruz Biotechnology, Germany), anti-mouse CD11b (Novus Biologicals, USA). Western blots were quantified with ImageJ software and normalized to the respective loading controls.

## Rap1 activation assay

Active Rap1 Pull-Down assay was performed according to the manufacturer's instructions (Fisher Scientific, Germany). Bone marrow-derived neutrophils were stimulated with CD95L (40 µg/ml) and the cell lysates were prepared. 100 µl Glutathione Resin and 20 µg of GST-RalGDS-RBD peptide were added to 500 µg lysate. GTPγS and GDP incubated lysates were used as positive and negative control respectively. After one-hour incubation at 4°C, resin beads were washed 4 times and then followed by incubation in 40 µl of 2x laemmli buffer at 95°C for 5 min. Precipitates were electrophoresed and blotted for anti-Rap1.

## Soluble ICAM1 binding assay

Soluble ICAM1 binding assay was performed as previously described (*Lefort et al., 2012*). Mouse bone marrow-derived neutrophils were suspended in Hanks Balanced Salt Solution containing 1 mM $CaCl2$ and $MgCl2$, and then the cell suspension was planted in CD95L-precoated chamber in the presence of ICAM1/FC (20 µg/ml, R&D systems, USA) and PE-conjugated anti-human IgG1 (Fc-specific; Southern Biotechnology, USA) for 5min at 37°C. Anti-CD11b (10 µg/ml) antibody was used to block the Mac-1-dependent ICAM1 binding. The binding of ICAM1 was determined by flow cytometry.

## Integrin activation reporter assay

Integrin conformational change upon CD95L treatment was tested by staining with reporter antibodies recognizing specific epitopes of integrin at different statuses. To test the binding, U937 cells (10 million/ml) were premixed with anti-Human CD11/CD18 (mab24) or anti-Human CD11/CD18 (KIM127) and perfused through the human E-selectin coated flow chamber with a syringe pump (New Era Pump Systems, USA) at the flow rate of 3 µl/min upon the stimulation with soluble CD95L (60 ng/ml) or immobilized CD95L (10 µg/ml for coating). The assembly and coating of the flow chamber was the same as described for the autoperfused mouse flow chamber assay. Cells flowed through the chamber were collected and fixed in 2% PFA. Then the fixed cells were stained with PE anti-mouse IgG and analyzed with flow cytometry.

## Proximity ligation assay and image analysis

Colocalization of integrin $\alpha_L$ to CD95 was determined by proximity ligation assay. dHL60 cells were planted in CD95L-precoated chamber for 10 min at 37°C. Following stimulation, cells were fixed with 2% PFA and stained with anti-CD11a (EP1285Y, abcam, UK) and anti-CD95 (APO-1-1, Enzo Life Science, Germany) antibodies. After washing, the proximity ligation assay was performed by using duolink kit following the manufacturer's instruction (Duolink In Situ Red Starter Kit Mouse/Rabbit, Sigma-Aldrich). Then the cells were stained with DAPI and Alexa 647-conjugated Phalloidin. Immunofluorescent signals were recorded with a TCS SP5 confocal microscope (Leica, Germany). Fluorescence images were analyzed with a programmed imageJ algorithm. The number of PLA and integrated density of PLA were accessed.

## Cecal ligation and puncture (CLP)-induced sepsis and bacterial culture

Sepsis was induced by CLP performed as previously described (*Rittirsch et al., 2009*). Age matched littermates of *Fas*<sup>f/f</sup> and *Fas*<sup>f/f</sup>::*Lyz2*<sup>Cre</sup> mice were used. Briefly, mice were anesthetized and cecum was exposed and ligated. To induce serve sepsis, the cecum was punctured twice with an 18-gauge needle, after which a small drop of feces was extruded from each puncture site to ensure patency. Mice were sacrificed 6 hr after CLP, at which time the peritoneal cavity was lavaged with 3 ml sterile phosphate-buffered saline containing 1 mM EDTA, and blood and spleen were collected. Aliquots of peritoneal lavage, blood and spleen homogenate were serially diluted, overnight cultured on sheep blood agar plates under 37°C, and then the number of CFUs was determined.

## Statistical evaluation

Statistical significance between groups were evaluated by one-way ANOVA using Bonferroni multiple comparison post hoc test for multiple groups comparison or Student's *t* test for two groups comparison. All data were presented as mean ± standard error of the mean (SEM) unless otherwise indicated. Statistical significance was determined by the p-value of the statistical test and deemed

as significant *p<0.05; strongly significant **p<0.01 and highly significant ***p<0.001. Statistical analysis was performed with GraphPad Prism (Version 5.01).

Significance for bacterial clearance experiments was evaluated by running a linear mixed model for the log-CFU with the random covariable of time point and the fixed covariable of gender, using SAS (Version 9.2).

## Acknowledgements

We thank Martina Schnölzer from the German Cancer Research Center (DKFZ) proteomics Core Facility for assistance with MS and Damir Krunic (Light Microscopy Facility, DKFZ) for assistance with PLA data analysis. We acknowledge Carsten Watzl (Leibniz Research Centre for Working Environment and Human Factors, Germany) for critical reading of the manuscript. This study was supported by the DKFZ. The authors declare no competing financial interests.

## Additional information

### Funding

| Funder | Author |
| --- | --- |
| Deutsches Krebsforschungs-zentrum | Ana Martin-Villalba |

The funders had no role in study design, data collection and interpretation, or the decision to submit the work for publication.

### Author contributions

LGa, Conception and design, Acquisition of data, Analysis and interpretation of data, Drafting or revising the article, Contributed unpublished essential data or reagents; GSG, LGo, HB, Acquisition of data, Analysis and interpretation of data, Drafting or revising the article; AZ, AM-V, Conception and design, Analysis and interpretation of data, Drafting or revising the article

### Author ORCIDs

Ana Martin-Villalba, http://orcid.org/0000-0002-9405-8910

### Ethics

Animal experimentation: Animal experiments in this study were performed in accordance with institutional guidelines of the German Cancer Research Center and were approved by the Regierungspräsidium Karlsruhe, Germany (Permit Number G188/13).

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
