## [Decision Letter]

Thank you for submitting your article "Endothelial cell-derived CD95 ligand serves as a chemokine in induction of neutrophil slow rolling and adhesion" for consideration by *eLife*. Your article has been favorably evaluated by Harry Dietz (Senior Editor) and three reviewers, one of whom is a member of our Board of Reviewing Editors. The following individuals involved in review of your submission have agreed to reveal their identity: Scott Simon (Reviewer #2); Tatiana V Byzova (Reviewer #3).

The reviewers have discussed the reviews with one another and the Reviewing Editor has drafted this decision to help you prepare a revised submission.

All three of the reviewers found the work interesting and potentially informative. However, all found major problems including 1) the need to address the state of CD95L in their system. Whether their soluble and surface-bound CD95L is monomeric or trimeric is not known and consequently, how it activates CD95 is not clear. While the full mechanism behind activation of integrins by CD95L2 need not be worked out in this paper, the details of their reagents must be provided in order to allow these experiments to be reproduced by others. 2) There are internal inconsistencies in the data, identified in the individual comments. These must be addressed. 3) The authors do not rule out other important modifiers or explanations such as the role of E-selectin. 4) It is not clear which (other) leukocyte integrins may be involved. After an online discussion, the reviewers thought that addressing these issues was necessary to validate their conclusions and therefore for acceptance of the manuscript. However, they thought that depending on the answers, additional experimentation to address these questions could be completed in a reasonable time frame if the authors chose to do so, and therefore, that they be given the opportunity to resubmit.

The full reviews are appended below for your consideration.

*Reviewer #1:*

The authors present evidence both in vitro and in vivo to show that interactions between CD95L on endothelial cells and CD95 on myeloid cells promotes slow rolling and adhesion by activation of β2 integrins through a pathway involving CD95, a src family kinase, Syk and/or Btk and PLCγ2. This is an interesting premise; however, there are other interpretations that are not ruled out and several points about the experimental system that require explanation.

The manuscript skips back and forth between experiments in mice using three different models (cremaster muscle stimulated with TNFα, thioglycollate peritonitis, and CLP) and uses various monocyte/macrophage cell lines (U937, fetal liver-derived macrophages) for each figure. I worry that the results may depend on the cell type and not be universal.

I could not find a description of the CD95L (FasL) that the authors use. CD95L is a transmembrane protein and acts as a homotrimer. How did the authors "make" CD95L? Was it commercially available? Did they use just the extracellular domain? (How) was it made trimeric? These are all important questions that need to be answered in order to interpret the data.

Figure 1. Why would i.v. injection of CD95L be expected to decrease rolling velocity of PMN? How would it activate CD95 if not trimeric and not membrane-anchored?

Figure 1. If the decreased adhesion and TEM of PMN is due to the role of CD95 in slow rolling, how do they explain the decreased adhesion and TEM under conditions in which slow rolling is not affected? This would imply some additional role for CD95 in adhesion and transmigration.

Figure 2. In the cremaster muscle in response to TNFα, PMN in endothelial cell-specific CD95L KO mice have and increased rolling velocity, while in Figure 1 myeloid-specific CD95 KO neutrophils have the same rolling velocity as wild-type in the same model. If these are ligands for each other, the same effect should be seen. Why is this not so?

Figure 4. The effects on integrin activation and ICAM-1 binding seem rather modest. There is no explanation of what the x-axis is in 4G.

Figure 5. The effect on bacterial clearance seems small. Mice were sacrificed at 6 hours, which seems to be a relatively short time point. What would happen if this were extended to 24 hours? What would be the effect on mortality? Other very important questions: How many PMN enter the peritoneal cavity under these conditions? One cannot extrapolate from the thioglycollate peritonitis experiments. The authors must measure PMN infiltration as well as determine whether the increased CFUs in the CD95 KO mice is due to defective PMN transmigration, defective PMN apoptosis, or defective bacterial phagocytosis and killing.

*Reviewer #2:*

Summary

The manuscript of Martin-Villalba et al. aims to show that CD95 mediates leukocyte slow rolling upon engagement with CD95L presented by endothelial cells. The authors show that presentation with CD95L results in phosphorylation of Syk/Btk/PLCy2/Rap1, an activation pathway that leads to integrin activation which in turn mediates slow rolling. Additionally, the authors show that CD95 in myeloid cells is required for bacterial clearance in a CLP-induced animal model of sepsis. The authors show a great deal of data collected through a thorough analysis of the effects of CD95L in vitro and in vivo. They determine that CD95L interaction results in activation of B2 integrin and subsequent decrease in rolling velocity and an increase in transendothelial migration that results in better bacterial clearance than in CD95-/- cells. However, the difference between CD95L signaling has not been decoupled from outside-in signaling induced by E-selectin engagement as the effect seems somewhat cooperative based on the data. Additionally, CD95L has often been associated with the induction of apoptosis and the authors must show that the cells are not moving towards an apoptotic phenotype when CD95 is engaged. All in all, the authors present a novel interaction between leukocytes and endothelial cells that results in increased recruitment via outside-in signaling. With some additional experiments and verifications of cell lifetime the paper would make a strong addition to the literature.

1) In Figure 1 the authors show a decrease in rolling velocity when presented with CD95L on the substrate or (to a lesser degree) with circulating CD95L. Using CD95 KO mouse cells, the decrease is abolished. However, the authors only show this on CD95L i.v. injected cells. What about with CD95L coating? That response shows a much larger decrease either implying additional receptor interactions that require shear force.

A) I believe it is important to include rolling velocity on E-selectin alone substrates to ensure that CD95L causes slow rolling through integrin activation.

B) An anti-Mac-1 mAB blocking should be compared with the anti-CD11a intervention in 1D.

C) An important parameter to confirm normalized PMN numbers would be circulating count for the various mouse lines, which are not discussed.

2) It is stated that tethering via E-selectin in the flow chamber is necessary for ligation and signaling via CD95L. However, is it possible to mediate this signaling via P-selectin or L-selectin and still observe CD95-signaled slow rolling? This would indicate that E-selectin specifically associates in the membrane at sites of rolling to arrest with CD95 and LFA-1 and should be addressed.

3) Increased extended and high affinity B2 integrin occur when cells are perfused with soluble CD95L but not with immobilized CD95L in Figure 4. This is contrary to what is shown in Figure 1 where coated CD95L induces slower rolling. The authors claim this is due to activation of B2 integrin, how is this dichotomy addressed?

A) Figure 4—figure supplement 1 also shows that soluble CD95L without E-selectin present shows no effect on KIM127 or mAb24 binding. The authors then make the claim that "CD95L treatment induces integrin conformational changes". This conclusion is not supported solely based on the data, it seems more likely that CD95L alone does not affect LFA-1 but rather enhances outside-in signaling via E-selectin, since without it no change is observed.

) Additionally in Figure 4 the cells stained with DAPI do not seem to exhibit a multi-lobed nucleus. Have these HL-60 cells truly been differentiated into PMN like cells? Additionally, the authors do not quantify F-actin concentration per cell. Based on the images provided I would say cells presented with CD95L have more phalloidin binding. This would imply a greater amount of actin polymerization and could help further the authors' claims.

4) The manuscript is very well put together but I am concerned that the CD95L is generally associated with induction of apoptosis. Is it possible these cells are being signaled toward apoptosis and that results in a cell membrane restructuring (increased activation, potential activation of integrin). It would be useful to show live/dead cell staining on your flow cytometry panels. It also may be important to observe long term cells survival to show that CD95L does not hasten programmed cell death of populations of PMN under study.

*Reviewer #3:*

This manuscript summarizes a series of reasonably novel and interesting results implicating CD95/CD95L on myeloid and endothelial cells, respectively, in the process of myeloid cell adhesion and transmigration into the sites of inflammation. The authors take an advantage of LysM-Cre and Cdh5-cre for myeloid and endothelial cells, respectively. These approaches provide comprehensive and conclusive results regarding the role of CD95/CD95L in slow rolling of myeloid cells. The authors also show the main signaling events involved in integrin activation by CD95, such as a complex formation between integrin, CD95, E-selectin etc. Overall, this is a solid and comprehensive study of substantial novelty.

I have several concerns which require clarification and additional experimentation.

The authors often refer to the analysis of neutrophils while using LysM-specific Cre as well as LysM-GFP. However, LysM is commonly known to be a general myeloid cell marker (myeloblasts, macrophages, and neutrophils) rather than strictly neutrophil marker. Caution is needed while dealing with this particular Cre. Another neutrophil relevant and, possibly more appropriate driver for Cre is Ly6G.

While considering the main integrin involved in rolling and adhesion for LysM-expressing cells, alphambeta2 appears to be more prominent. In fact, there is a discrepancy in the manuscript regarding the main integrin (alphalbeta2 on 5; alphambeta 2 (CD11b) was used as a marker for gaiting in Figure 2). In fact, Figure 2 takes an advantage of two best markers for neutrophils gating. For analysis of integrin expression and activation I suggest similar gating of GFP-positive cells and analysis of neutrophils separately from monocytes, using activation reporter abs to both integrins.

Since the authors claim that CD95 signaling leads to integrin activation, they need to demonstrate this directly by a combination of methods, not merely binding of KIM and mAb24. For these two abs, please show FACs profiles and both,% of positive cells as well as MFI. Why the authors use anti-CD11b abs? Both integrins (alphaL and alpham beta2) are expected to be activated by CD95 pathway. In fact, other integrin (i.e. beta3) are likely to be activated as well. Do the authors claim this to be CD11a specific? If this is the case, the authors have to analyze all integrins separately. It will help to show impaired β 2 integrin activation on CD95 KO cells in the presence and absence of CD95L. Second, perhaps more difficult aspect is to show impaired integrin activation on CD95L KO endothelium.

---

## [Author Response]

[…]

*Reviewer #1:*

*The authors present evidence both* in vitro *and* in vivo *to show that interactions between CD95L on endothelial cells and CD95 on myeloid cells promotes slow rolling and adhesion by activation of β2 integrins through a pathway involving CD95, a src family kinase, Syk and/or Btk and PLCγ2. This is an interesting premise; however, there are other interpretations that are not ruled out and several points about the experimental system that require explanation.*

*The manuscript skips back and forth between experiments in mice using three different models (cremaster muscle stimulated with TNFα, thioglycollate peritonitis, and CLP) and uses various monocyte/macrophage cell lines (U937, fetal liver-derived macrophages) for each figure. I worry that the results may depend on the cell type and not be universal.*

We applied TNFa-induced inflammation model of cremaster muscle to study the parameters of CD95-induced neutrophil slow rolling, adhesion and trans-endothelial migration under sterile inflammation conditions. Thioglycollate-induced peritonitis is also a sterile inflammation model, while it is more related to pathological conditions. In accordance with our previous study (Letellier et al., 2010), our data from these two models show that CD95 signaling mediates neutrophil recruitment in sterile inflammation. In order to examine whether CD95-mediated recruitment is a general mechanism during inflammation, we applied CLP model which is a pathogen-induced systemic inflammation model and our data shows that CD95 is also involved in mediating neutrophil recruitment in pathogenic inflammation (see revised Figure 5, also see subsection “CD95 in myeloid cells is required for bacterial clearance”). We cultured macrophages from fetal liver cells in order to get enough cells for the signaling study as *Syk* deficiency (*Syk-/-*) is perinatal-lethal in mice. Human-derived U937 cell line was used in the integrin activation assay as the antibodies used in these assays were only human antigen specific.

*I could not find a description of the CD95L (FasL) that the authors use. CD95L is a transmembrane protein and acts as a homotrimer. How did the authors "make" CD95L? Was it commercially available? Did they use just the extracellular domain? (How) was it made trimeric? These are all important questions that need to be answered in order to interpret the data.*

The CD95L we used is a fusion protein of trimeric human CD95L-receptor binding domain fused with T4- Foldon motif from the fibritin of the bacteriophage T4 (CD95L-T4) and purified from CD95L-T4 plasmid-transfected HEK293T cells (see details in Kleber et al., 2008). It is commercially available from IBA GmbH, Göttingen, Germany. Also see the subsection “Autoperfused mouse flow chamber assay”.

*Figure 1. Why would i.v. injection of CD95L be expected to decrease rolling velocity of PMN? How would it activate CD95 if not trimeric and not membrane-anchored?*

See above.

*Figure 1. If the decreased adhesion and TEM of PMN is due to the role of CD95 in slow rolling, how do they explain the decreased adhesion and TEM under conditions in which slow rolling is not affected? This would imply some additional role for CD95 in adhesion and transmigration.*

We agree that CD95 is also involved in modulating adhesion and transmigration. And actually we have previously reported that CD95 induces myeloid cells transmigration via activating the Syk-PI3K-MMP9 pathway (Letellier et al., 2010). These findings are already cited in the Introduction (second paragraph).

*Figure 2. In the cremaster muscle in response to TNFα, PMN in endothelial cell-specific CD95L KO mice have and increased rolling velocity, while in Figure 1 myeloid-specific CD95 KO neutrophils have the same rolling velocity as wild-type in the same model. If these are ligands for each other, the same effect should be seen. Why is this not so?*

We speculated in the previous version of the manuscript that this effect might be due to the redundant function of TNF-α and CD95 in regulating slow rolling. Interestingly, two studies showed that TNF is involved in neutrophil and T-cell adhesion via TNF-induced integrin inside-out signaling (Lauterbach et al., 2008; Li et al., 2016). In the revised version, we demonstrate that neutrophils from naïve *Fas^<f/f>^::Lyz2^<cre>^*mice express higher level of TNFR2 but similar levels of TNFR1. However, at 6h after CLP, neutrophils from *Fas^<f/f>^::Lyz2^<cre>^*mice express higher TNFR1 but similar level TNFR2 (see revised Figure 1—figure supplement 2). As TNF-a was applied to induce the inflammation in the cremaster muscle, the compensated effect of CD95 deficiency on slow rolling might be due to the up-regulated TNFR in neutrophils from *Fas^<f/f>^::Lyz2^<cre>^*mice. Also see the fourth paragraph of the subsection “CD95L stimulation induces neutrophil slow rolling”.

*Figure 4. The effects on integrin activation and ICAM-1 binding seem rather modest. There is no explanation of what the x-axis is in 4G.*

Similar ‘modest’ binding of integrin activation reporter antibodies or ICAM1 has also been shown in other publication, such as (Figure 3 of Pruenster et al., 2015,) and (Figure 6C of Germena et al., 2015,). We assume it might be due to the basal activation of the cells we used or simply because the effect of CD95L is not as robust as chemokines/cytokines, such as CXCL1 in activating integrin.

Figure 4 demonstrates the number of PLA signal in a violin plot in control or CD95L-treated dHL60 cells. X-axis in Figure 4 shows control or CD95L treatment.

*Figure 5. The effect on bacterial clearance seems small. Mice were sacrificed at 6 hours, which seems to be a relatively short time point. What would happen if this were extended to 24 hours? What would be the effect on mortality? Other very important questions: How many PMN enter the peritoneal cavity under these conditions? One cannot extrapolate from the thioglycollate peritonitis experiments. The authors must measure PMN infiltration as well as determine whether the increased CFUs in the CD95 KO mice is due to defective PMN transmigration, defective PMN apoptosis, or defective bacterial phagocytosis and killing.*

In our preliminary experiment of peritonitis we observed that neutrophils peaked in peritoneal cavity 6h after thioglycollate injection (data not shown). Another study also showed that in the CLP model neutrophils peaked in peritoneal cavity at 6h after CLP (Deng et al., 2013). Therefore, we chose 6h post CLP as the time point to study the effect of CD95 on neutrophil’s recruitment, which could be reflected by the difference of CFUs numbers between wt and CD95 KO mice.

In the revised version we have added the data of PMN infiltration 6h after CLP. *Fas^<f/f>^::Lyz2^<cre>^* mice shows significantly less PMN infiltration compared to the control mice (revised Figure 5). Thus, altogether these data indicate that the impaired bacterial clearance in *Fas^<f/f>^::Lyz2^<cre>^*mice is at least partially due to the reduced infiltration of neutrophils. The additional impact of CD95 deficiency on neutrophil activity should be clarified in future studies.

Also see the subsection “CD95 in myeloid cells is required for bacterial clearance” and the last paragraph of the Discussion.

*Reviewer #2:*

*Summary*

*The manuscript of Martin-Villalba et al. aims to show that CD95 mediates leukocyte slow rolling upon engagement with CD95L presented by endothelial cells. The authors show that presentation with CD95L results in phosphorylation of Syk/Btk/PLCy2/Rap1, an activation pathway that leads to integrin activation which in turn mediates slow rolling. Additionally, the authors show that CD95 in myeloid cells is required for bacterial clearance in a CLP-induced animal model of sepsis. The authors show a great deal of data collected through a thorough analysis of the effects of CD95L* in vitro *and* in vivo*. They determine that CD95L interaction results in activation of B2 integrin and subsequent decrease in rolling velocity and an increase in transendothelial migration that results in better bacterial clearance than in CD95-/- cells. However, the difference between CD95L signaling has not been decoupled from outside-in signaling induced by E-selectin engagement as the effect seems somewhat cooperative based on the data. Additionally, CD95L has often been associated with the induction of apoptosis and the authors must show that the cells are not moving towards an apoptotic phenotype when CD95 is engaged. All in all, the authors present a novel interaction between leukocytes and endothelial cells that results in increased recruitment via outside-in signaling. With some additional experiments and verifications of cell lifetime the paper would make a strong addition to the literature.*

*1) In Figure 1 the authors show a decrease in rolling velocity when presented with CD95L on the substrate or (to a lesser degree) with circulating CD95L. Using CD95 KO mouse cells, the decrease is abolished. However, the authors only show this on CD95L i.v. injected cells. What about with CD95L coating? That response shows a much larger decrease either implying additional receptor interactions that require shear force.*

Immobilized CD95L-induced slow rolling was also abolished in *Fas^<f/f>^::Lyz2^<cre>^*mice (see revised Figure 1 and also subsection “CD95L stimulation induces neutrophil slow rolling”, first paragraph).

*A) I believe it is important to include rolling velocity on E-selectin alone substrates to ensure that CD95L causes slow rolling through integrin activation.*

*B) An anti-Mac-1 mAB blocking should be compared with the anti-CD11a intervention in 1D.*

We have added this piece of data to the revised version of the manuscript.

Anti-MAC-1 (CD11b) mab didn’t block the effect of CD95L on slow rolling (see revised Figure 1—figure supplement 1 and also subsection “CD95L stimulation induces neutrophil slow rolling”, second paragraph).

*C) An important parameter to confirm normalized PMN numbers would be circulating count for the various mouse lines, which are not discussed.*

Absolute numbers of circulating PMN were added to the revised version. See revised Figure 1—figure supplement 3, Figure 2—figure supplement 2 and also subsection “CD95L stimulation induces neutrophil slow rolling”, last paragraph and subsection “Endothelial cells-derived CD95L mediates neutrophil recruitment”, last paragraph.

*2) It is stated that tethering via E-selectin in the flow chamber is necessary for ligation and signaling via CD95L. However, is it possible to mediate this signaling via P-selectin or L-selectin and still observe CD95-signaled slow rolling? This would indicate that E-selectin specifically associates in the membrane at sites of rolling to arrest with CD95 and LFA-1 and should be addressed.*

CD95L shows no effect on rolling velocity of neutrophil on L-selectin/P-selectin and ICAM1 coated chamber (see revised Figure 1—figure supplement 1 and also subsection “CD95L stimulation induces neutrophil slow rolling”, third paragraph). It suggests that the CD95 specifically modulates E-selectin-associated slow rolling.

*3) Increased extended and high affinity B2 integrin occur when cells are perfused with soluble CD95L but not with immobilized CD95L in Figure 4. This is contrary to what is shown in Figure 1 where coated CD95L induces slower rolling. The authors claim this is due to activation of B2 integrin, how is this dichotomy addressed?*

Immobilized CD95L also increased the binding of reporter antibodies of extended and high affinity integrin, but just not as effective as the soluble CD95L treated one (Figure 4). It might due to the coating concentration of CD95L, which did not meet the optimized concentration for human cell lines as compared to the concentration used for mouse cells in Figure 1.

*A) Figure 4—figure supplement 1 also shows that soluble CD95L without E-selectin present shows no effect on KIM127 or mAb24 binding. The authors then make the claim that "CD95L treatment induces integrin conformational changes". This conclusion is not supported solely based on the data, it seems more likely that CD95L alone does not affect LFA-1 but rather enhances outside-in signaling via E-selectin, since without it no change is observed.*

We agree that the CD95-induced conformational changes are E-selectin-dependent. See the second paragraph of the subsection “CD95L stimulation activates integrin”.

*B) Additionally in Figure 4 the cells stained with DAPI do not seem to exhibit a multi-lobed nucleus. Have these HL-60 cells truly been differentiated into PMN like cells? Additionally, the authors do not quantify F-actin concentration per cell. Based on the images provided I would say cells presented with CD95L have more phalloidin binding. This would imply a greater amount of actin polymerization and could help further the authors' claims.*

In this study, HL-60 cells were allowed to differentiate in presence of 1.3% DMSO for 6 days before used. The DAPI staining pictures in Figure 4 were derived from maximal projections of multi stacks pictures, therefore they lost some information of the nucleus morphology. The following pictures show the selected stacks of DAPI staining for cells showed in Figure 4. The nucleuses are non-spherical and exhibit as lobed structure.

Author response image 1.**DOI:**
http://dx.doi.org/10.7554/eLife.18542.014

It is a great idea to quantify the F-actin polymerization in control and CD95L-treated cells, since it further supports the data showing activation of integrin upon CD95L treatment. There was more phalloidin binding in CD95L-treated cells as compared to control cells, however it is not significantly different with the limited number of cells we examined (see revised Figure 4—figure supplement 1 and also subsection “CD95 recruits and associates with integrin”, first paragraph).

*4) The manuscript is very well put together but I am concerned that the CD95L is generally associated with induction of apoptosis. Is it possible these cells are being signaled toward apoptosis and that results in a cell membrane restructuring (increased activation, potential activation of integrin). It would be useful to show live/dead cell staining on your flow cytometry panels. It also may be important to observe long term cells survival to show that CD95L does not hasten programmed cell death of populations of PMN under study.*

In our previous study we have shown that CD95L stimulation had no effect on inducing apoptosis at the concentration we applied to the myeloid cells (Letellier et al., 2010). Neutrophils and macrophages stimulated with CD95L also showed no recruitment of Fas-Associated Death Domain (FADD) to the Death Domain (DD) of CD95.

Accumulating evidence demonstrates that CD95 also has important non-apoptotic functions, such as mediating cell survival, proliferation and migration, mostly through the activation of three major MAPKs, c-JUN N-terminal kinase (*JNK*), p38 and extracellular signal-regulated kinase (ERK), NF- κB and PI3K pathways (reviewed by Wajant et al., 2003; Peter et al., 2007; Martin-Villalba et al., 2013). Interestingly, a recent paper (Poissonnier et al., 2016) reported that CD95 signaling promoted Th17 cell transmigration, instead of inducing apoptosis, which shows a similar function in myeloid cells as we previously observed (Letellier et al., 2010). These data are cited in the manuscript.

*Reviewer #3:*

[…]

*I have several concerns which require clarification and additional experimentation.*

*The authors often refer to the analysis of neutrophils while using LysM-specific Cre as well as LysM-GFP. However, LysM is commonly known to be a general myeloid cell marker (myeloblasts, macrophages, and neutrophils) rather than strictly neutrophil marker. Caution is needed while dealing with this particular Cre. Another neutrophil relevant and, possibly more appropriate driver for Cre is Ly6G.*

We agree that the Ly6Gcre would be a better mouse line for this study. And we are also aware that *lysozyme M* gene is expressed by myeloid lineage cells and didn’t claim that we use *Lyz2^<Cre>^* for neutrophil-specific KO. In the autoperfused flow chamber assay, using the *Lyz2^<CreGFP>^* reporter mice, 89 ± 2% of the rolling cells were brightly fluorescent (Chesnutt et al., 2006). The purity of neutrophils among the rolling cells had also been identified by staining of myeloperoxidase of fixed cells in the flow chambers (Zarbock et al., 2007). For the model of inflamed cremaster muscles, it’s been reported that >95% of all adherent and rolling leukocytes are neutrophils (Jung et al., 1998, also see subsection "CD95L stimulation induces neutrophil slow rolling", fourth paragraph).

*While considering the main integrin involved in rolling and adhesion for LysM-expressing cells, alphambeta2 appears to be more prominent. In fact, there is a discrepancy in the manuscript regarding the main integrin (alphalbeta2 on 5; alphambeta 2 (CD11b) was used as a marker for gaiting in Figure 2). In fact, Figure 2 takes an advantage of two best markers for neutrophils gating. For analysis of integrin expression and activation I suggest similar gating of GFP-positive cells and analysis of neutrophils separately from monocytes, using activation reporter abs to both integrins.*

A previous study showed that antibody against LFA-1 (CD11a, integrin αL) could block the E-selectin/ICAM1-induced slow rolling, while antibody against MAC-1 (CD11b, integrin αM) could not (Zarbock et al., 2007). And our data show that CD11a antibody, but not CD11b antibody, also block CD95L-induced slow rolling (see revised version Figure 1, Figure 1—figure supplement 1 and also see the subsection “CD95L stimulation induces neutrophil slow rolling”, second paragraph), which indicates that CD95 signaling induces slow rolling via activating integrin alphaL.

For the studies of integrin expression level, all the analysis was based on the gating of Ly6G high and CD11b positive neutrophils of which monocytes were excluded. We didn’t use GFP-reporter mouse here. Available integrin activation reporter abs are only specific for human antigens, thus we used only human cell lines for this study.

*Since the authors claim that CD95 signaling leads to integrin activation, they need to demonstrate this directly by a combination of methods, not merely binding of KIM and mAb24. For these two abs, please show FACs profiles and both,% of positive cells as well as MFI. Why the authors use anti-CD11b abs? Both integrins (alphaL and alpham beta2) are expected to be activated by CD95 pathway. In fact, other integrin (i.e. beta3) are likely to be activated as well. Do the authors claim this to be CD11a specific? If this is the case, the authors have to analyze all integrins separately. It will help to show impaired β 2 integrin activation on CD95 KO cells in the presence and absence of CD95L. Second, perhaps more difficult aspect is to show impaired integrin activation on CD95L KO endothelium.*

We used the integrin activation reporter antibody binding assay and soluble ICAM1 binding assay to directly prove that CD95 signaling activates integrin (Figure 4). And we show that the intermediate factor for integrin activation- the Rap1GTPase was also activated upon CD95L stimulation.

FACS profiles for KIM127 and mAb24 staining are added, see revised version Figure 4—figure supplement 1 and also see the second paragraph of the subsection “CD95L stimulation activates integrin”. MFI was measured by gating on the whole population. Anti-CD11b abs here was used to block the binding of reporter antibodies, which could be induced by CD11b activation.

According to our data, CD95-induced slow rolling is CD11a specific as it can be blocked by CD11a (integrin αL) antibody but not CD11b (integrin αM) antibody (see Figure 1, Figure 1—figure supplement 1 and also the subsection “CD95L stimulation induces neutrophil slow rolling”, second paragraph). The rest known integrin α subunits binding to integrin β2 are integrin αX which is not expressed by neutrophils, and integrin αD which ligates to ICAM3/VCAM1. Thus, we don’t think that further blocking integrin β2 (CD18) or block integrin activation on CD95L KO endothelial cells is critical for this study.